# Explaining Preferences with Shapley Values

**Robert Hu**[*†]
Amazon
London

**Siu Lun Chau**[*]
Department of Statistics
University of Oxford

**Jaime Ferrando Huertas**
Shaped
New York

**Dino Sejdinovic** [†]
School of Computer and Mathematical Sciences
University of Adelaide

## Abstract

While preference modelling is becoming one of the pillars of machine learning, the problem of preference explanation remains challenging and underexplored. In this paper, we propose PREF-SHAP, a Shapley value-based model explanation framework for pairwise comparison data. We derive the appropriate value functions for preference models and further extend the framework to model and explain *context specific* information, such as the surface type in a tennis game. To demonstrate the utility of PREF-SHAP, we apply our method to a variety of synthetic and real-world datasets and show that richer and more insightful explanations can be obtained over the baseline.

## 1 Introduction

Preference learning [1] is a classical problem in machine learning, where one is interested in learning the order relations on a collection of data items. Preference learning algorithms [2–5] often assume that there is a latent utility function $f : \mathcal{X} \mapsto \mathbb{R}$ dictating the outcome of preferences, where $\mathcal{X}$ denotes the domain of item covariates. An explicit feedback such as item ratings or rankings from recommender systems can be treated as noisy evaluations of $f$, whereas pairwise comparison data (also known as duelling data) arising from, e.g., sports match outcomes [6, 7] can be used to implicitly infer $f$, i.e. item $\mathbf{x}^{(\ell)}$ is preferred over (beats) item $\mathbf{x}^{(r)}$ when $f(\mathbf{x}^{(\ell)}) > f(\mathbf{x}^{(r)})$. As shown by Kahneman and Tversky [8], humans often struggle with evaluating absolute quantities when it comes to eliciting preferences, but are broadly capable of evaluating relative differences, a core observation often exploited in preference learning. Motivated by such, this work will focus on explaining preferences inferred using duelling data.

Explaining preference models is crucial when they are applied in areas such as recommendation systems [9], finance [10], and sports science [11] for the practitioner to trust, debug and understand the value of their findings [12]. However, despite its importance, no prior work has studied this problem to the best of our knowledge. While one may suggest applying existing explainability tools such as LIME [13], or SHAP [14] to a learned utility function $f$, we reason that this approach only explains the utility but not the mechanism of eliciting preferences itself. We highlight the important differences between these two viewpoints in our numerical experiments. Moreover, the utility-based model places a strong *rankability* assumption on the underlying preferences, meaning that if we define $\mathbf{x}^{(\ell)} \preceq \mathbf{x}^{(r)} \iff f(\mathbf{x}^{(\ell)}) \leq f(\mathbf{x}^{(r)})$, then $\preceq$ is a total order on all the items. However, as Pahikkala et al. [15] and Chau et al. [16] have discussed, there are many departures from rankability in practice, e.g. we might easily see a preference of $A$ over $B$, $B$ over $C$, but $C$ over $A$ – conforming to the *rock-paper-scissors* relation. Such inconsistent preferences are under frequent study in social

---

[*]Equal contribution, order decided by coinflip

[†]Work mainly done while the authors were with the Department of Statistics, University of Oxford

36th Conference on Neural Information Processing Systems (NeurIPS 2022).

choice theory [17, 18], and are of wider interest in both healthcare [19] and retail [20] where data are both large and noisy.

To move beyond the rankability assumption, we will utilise the *Generalised Preferential Kernel* from [16] to model the underlying preferences, and develop PREF-SHAP, a novel Shapley value [21]-based explainability toolbox, to explain the inferred preferences. Our contributions can be summarised as follows:

1. We propose PREF-SHAP, a novel Shapley value-based explainability algorithm, to explain preferences based on duelling data.

2. We empirically demonstrate that PREF-SHAP gives more informative explanations compared to the naive approach of applying SHAP to the inferred utility function $f$.

3. We release a high-performant implementation of PREF-SHAP at [22].

## 2   Background materials

We will first give a brief overview of preference learning and Shapley Additive Explanations (SHAP) [14], which are the two core concepts of our contribution, PREF-SHAP, described in Section 3.

**Notation**     Scalars are denoted by lower case letters, while vectors and matrices are denoted by bold lower case and upper case letters, respectively. Random variables are denoted by upper case letters. $\mathcal{X} \subseteq \mathbb{R}^d$ denotes the item space with $d$ features and $\mathcal{Y} = \{-1, 1\}$ is the binary preference outcome space[3]. We let $k : \mathcal{X} \times \mathcal{X} \to \mathbb{R}$ be a kernel function and $\mathcal{H}_k$ the corresponding reproducing kernel Hilbert space (RKHS).

### 2.1   Preference Learning

In this section, we will introduce the two approaches to model preferences from duelling data, namely the *utility based approach* and the more general approach from Chau et al. [16]. Formally, a preference feedback is denoted as *duelling*, when a pair of items $(\mathbf{x}^{(\ell)}, \mathbf{x}^{(r)}) \in \mathcal{X} \times \mathcal{X}$ is given to a user, and a binary outcome $y \in \mathcal{Y}$ telling us whether $\mathbf{x}^{(\ell)}$ or $\mathbf{x}^{(r)}$ won the duel, is observed. In general, we observe $m$ binary preferences among $n$ items, giving the data $D = \left(\mathbf{y}, \mathbf{X}^{(\ell)}, \mathbf{X}^{(r)}\right) = \left\{ (y_j, \mathbf{x}_j^{(\ell)}, \mathbf{x}_j^{(r)}) \right\}_{j=1}^m$. We also use $\mathbf{X} \in \mathbb{R}^{n \times d}$ to denote the full item covariate matrix.

**Utility-based Preference model (UPM)**     The following likelihood model is often used [2–5, 7] to model duelling feedback using a latent utility function $f$:

$$p\left(y \mid \mathbf{x}^{(\ell)}, \mathbf{x}^{(r)}\right) = \sigma\left(y\left(f\left(\mathbf{x}^{(\ell)}\right) - f\left(\mathbf{x}^{(r)}\right)\right)\right), \tag{1}$$

where $\sigma$ is the logistic CDF, i.e. $\sigma(z) = (1 + \exp(-z))^{-1}$. Maximum likelihood approaches are then deployed to learn the latent utility function $f$. Consequently, preferences between items can be inferred accordingly from $\mathbf{f} = \{f(\mathbf{x}_i)\}_{i=1}^n$, i.e. $\mathbf{x}_i$ is on average preferred over $\mathbf{x}_j$ if $\mathbf{f}_i \geq \mathbf{f}_j$.

Albeit elegant, there are several drawbacks to this approach in modelling preferences. As mentioned, using a one-dimensional vector $\mathbf{f}$ to derive preferences assumes that the items $\{\mathbf{x}_i\}_{i=1}^n$ are perfectly rankable, i.e. there is a total ordering on $\mathcal{X}$ which the true preferences are consistent with. This is a strong assumption that often does not hold in practice. For example, it is well studied that cognitive biases often lead to inconsistent human preferences in behavioural economics [8]. Moreover, the ranking community has also challenged this assumption by devising rankability metrics [23, 24] to test this restrictive assumption in practice.

**Generalised Preference Model (GPM)**     Chau et al. [16] proposed to model preference directly using a more general $g : \mathcal{X} \times \mathcal{X} \to \mathbb{R}$ that captures the preference within any pair of items, using the likelihood

$$p\left(y \mid \mathbf{x}^{(\ell)}, \mathbf{x}^{(r)}\right) = \sigma\left(yg\left(\mathbf{x}^{(\ell)}, \mathbf{x}^{(r)}\right)\right). \tag{2}$$

We note that $g$ has to be a skew-symmetric function to ensure the natural property $p(y \mid \mathbf{x}^{(\ell)}, \mathbf{x}^{(r)}) = 1 - p(y \mid \mathbf{x}^{(r)}, \mathbf{x}^{(\ell)})$. The utility based approach can be obtained as a special case of this model, i.e.

---

[3]Thus, we do not model 'draws' in match outcomes, but the model can be straightforwardly extended to include them by specifying the appropriate likelihood function.

by setting $g(\mathbf{x}^{(\ell)}, \mathbf{x}^{(r)}) = f(\mathbf{x}^{(\ell)}) - f(\mathbf{x}^{(r)})$. We propose that when one is interested in modelling (and thus explaining) pairwise preferences, we should consider the preference function $g$ directly instead of explaining preferences based on a restrictive utility model $f$.

We follow Chau et al. [16]'s approach to model $g$ non-parametrically using kernel methods [25]. We assume $g$ as a function lives in the following RKHS of skew-symmetric functions: given kernel $k : \mathcal{X} \times \mathcal{X} \to \mathbb{R}$ defined on the item space $\mathcal{X}$, the *generalised preferential kernel* $k_E$ on $\mathcal{X} \times \mathcal{X}$ is constructed as follows:

$$k_E\left(\left(\mathbf{x}_i^{(\ell)}, \mathbf{x}_i^{(r)}\right), \left(\mathbf{x}_j^{(\ell)}, \mathbf{x}_j^{(r)}\right)\right) = k\left(\mathbf{x}_i^{(\ell)}, \mathbf{x}_j^{(\ell)}\right) k\left(\mathbf{x}_i^{(r)}, \mathbf{x}_j^{(r)}\right) - k\left(\mathbf{x}_i^{(\ell)}, \mathbf{x}_j^{(r)}\right) k\left(\mathbf{x}_i^{(r)}, \mathbf{x}_j^{(\ell)}\right).$$

This kernel allows us to model the similarity across pairs of items. Moreover, if $k$ is a universal kernel [26], then $k_E$ also satisfies the corresponding notion of universality, meaning that the corresponding RKHS $\mathcal{H}_{k_E}$ is rich enough to approximate any bounded continuous skew-symmetric function arbitrarily well [16, Theorem. 1]. To infer $g \in \mathcal{H}_{k_E}$ using likelihood (2), one simply runs kernel logistic regression with data $\mathbf{y}$ as labels and $\left(\mathbf{X}^{(\ell)}, \mathbf{X}^{(r)}\right)$ as inputs. We will refer to this approach as the *Generalised Preference Model* (GPM).

We emphasize that *explaining* GPM allows us to specifically explain *inconsistent preferences*, which in contrast to *explaining rank* allows us to infer preferences even when transitivity is violated. Such insights can be of great importance in broader contexts such as decision theory [27] and utility theory [28] where transitivity does not hold.

**Incorporating context variables.** Besides item-level covariates $\mathbf{x} \in \mathcal{X}$, when there exist additional *context covariates* $\mathbf{u} \in \mathcal{U} \subseteq \mathbb{R}^{d'}$ that describe the context in which a specific pairwise comparison is made, they can be incorporated into the kernel design as discussed in Chau et al. [16, Appendix. B]. Examples of such context covariates could be court type when a tennis match is conducted, or where a different user compares two clothing items in e-commerce. Considering the enriched dataset $D = \left\{\left(y_j, \mathbf{u}_j, \mathbf{x}_j^{(\ell)}, \mathbf{x}_j^{(r)}\right)\right\}_{j=1}^m$, we can now model the preference incorporating the context as: $p(y \mid \mathbf{u}, \mathbf{x}^{(\ell)}, \mathbf{x}^{(r)}) = \sigma\left(g_U\left(\mathbf{u}, \mathbf{x}^{(\ell)}, \mathbf{x}^{(r)}\right)\right)$. Now, given a kernel $k_U$ defined on the context space $\mathcal{U}$, the context-specific preference function $g_U : \mathcal{U} \times \mathcal{X} \times \mathcal{X} \to \mathbb{R}$ can be learnt non-parametrically with the following kernel,

$$k_E^{(U)}\left(\left(\mathbf{u}_i, \mathbf{x}_i^{(\ell)}, \mathbf{x}_i^{(r)}\right), \left(\mathbf{u}_j, \mathbf{x}_j^{(\ell)}, \mathbf{x}_j^{(r)}\right)\right) = k_U\left(\mathbf{u}_i, \mathbf{u}_j\right) k_E\left(\left(\mathbf{x}_i^{(\ell)}, \mathbf{x}_i^{(r)}\right), \left(\mathbf{x}_j^{(\ell)}, \mathbf{x}_j^{(r)}\right)\right).$$

We refer to this approach as the *Context-specific Generalised Preference Model* (C-GPM).

## 2.2 Shapley Additive Explanations (SHAP)

To explain preferences, we will utilise the popular SHAP (SHapley Additive exPlanations) paradigm, which is based on the concept of Shapley values (SV). SV [21] were originally proposed as a credit allocation scheme for a group of $d$ players in the context of cooperative games, which are characterised by a value function $\nu : [0, 1]^d \to \mathbb{R}$ that measures *utility* of subsets of players. Formally, the Shapley value for player $j$ in game $\nu$ is defined as:

$$\phi_j(\nu) = \sum_{S \subseteq \Omega \setminus \{j\}} (|S|!(d - |S| - 1)!/d!) \left(\nu(S \cup j) - \nu(S)\right), \tag{3}$$

where $\Omega = \{1, ..., d\}$ is the set of players of the game. Given a value function $\nu$, the Shapley values are proven to be the only credit allocation scheme that satisfies a particular set of favourable and fair game theoretical axioms, commonly known as *efficiency*, *null player property*, *symmetry* and *additivity* [21]. Štrumbelj and Kononenko [29] later connect Shapley values to the field of *explainable machine learning* by drawing an analogy between model fitting and cooperative game. Given a specific data point, by considering its *features* as *players* participating in a game that measures features' utilities, the Shapley values obtained can be treated as *local feature importance scores*. Such games are typically defined through the value functions defined below.

**Definition 2.1** (Value functions). *Let $X$ be a random variable on $\mathcal{X} \subseteq \mathbb{R}^d$ and $f : \mathcal{X} \to \mathbb{R}$ a model from hypothesis space $\mathcal{H}$. The value function $\nu : \mathcal{X} \times [0, 1]^d \times \mathcal{H}$ is given by*

$$\nu_{\mathbf{x}, S}(f) = \mathbb{E}_{r(X_{S^c}|X_S = \mathbf{x}_S)} \left[f(\{X_S, X_{S^c}\}) \mid X_S = \mathbf{x}_S\right] \tag{4}$$

*where $r$ is an appropriate reference distribution, $X_S$ is the subvector of $X$ corresponding to the feature set $S$, $S^c$ is the complement of the feature set $S$ and $\{X_S, X_{S^c}\} = X$ denotes the concatenation of $X_S$ and $X_{S^c}$.*

In other words, given a data point $\mathbf{x}$, the utility of the feature subset $S$ is defined as the impact on the model prediction, after "removing" the contribution from $S^c$ via integration with respect to the reference distribution $r$. These "removal-based" strategies are common in the explainability literature [30]. Nonetheless, the correct choice of the reference distribution has been a long-standing debate [31]. Janzing et al. [32] argued from a causality perspective that the feature marginal distribution should be used as the reference distribution, i.e. $r(X_{S^c} \mid X_S = x_S) = p(X_{S^c})$ where $p$ is the data distribution. On the other hand, Frye et al. [33] disagreed by pointing out these "marginal" value functions ignore feature correlations and lead to unintelligible explanations in higher-dimensional data, and they instead advocate the use of conditional distribution as reference, i.e. $r(X_{S^c} \mid X_S = x_S) = p(X_{S^c} \mid X_S = x_S)$. Thus, there is no consensus and in fact, Chen et al. [31] took a neutral stand and argued the choice depends on the application at hand. This also leads to design of value functions for specific problems, e.g. improving local estimation [34], incorporating causal knowledge [35, 36] and modelling structured data [37]. In this paper, we will design an appropriate value function for preference learning and show that naive application of the existing value function to preference learning will lead to unintuitive results.

**Shapley value estimation.** Given a data point $\mathbf{x}$ and a model $f$, estimating Shapley values consist of two main steps: Firstly, for each feature subset $S \subseteq \Omega$, estimate the value function $\nu_{\mathbf{x},f}(S)$ either by Monte Carlo sampling from the reference distributions $r$, or by utilising a model specific structure to speed up the estimation such as in LINEARSHAP [29], DEEPSHAP [14], TREESHAP [38], and RKHS-SHAP [12]. The former sampling procedure is straightforward when $r$ is the marginal distribution, but computationally heavy and difficult when $r$ is the conditional distribution, as it involves estimating an exponential number of conditional densities [39]. Finally, after estimating the value functions, one can compute the Shapley values based on Eq. 3 or by utilising the efficient weighted least square approach proposed by Lundberg and Lee [14].

**Estimating value functions when $f \in \mathcal{H}_k$.** We give a review to the recently introduced RKHS-SHAP algorithm proposed by Chau et al. [12] as it is another core component for PREF-SHAP. RKHS-SHAP is a SV estimation method for functions in a given RKHS. It circumvents the need for any density estimation and utilises the arsenal of kernel mean embeddings [40] to estimate the value functions non-parametrically. Assume $k$ takes a product kernel structure across dimensions, then for any $f \in \mathcal{H}_k$, by applying the *reproducing property* [25], the value function can be decomposed as:

$$\nu_{\mathbf{x},S}(f) = \left\langle f, \mathbb{E}_{r(X_{S^c} \mid X_S = \mathbf{x}_S)} \left[ k \left( \{X_S, X_{S^c}\}, \cdot \right) \mid X_S = \mathbf{x}_S \right] \right\rangle_{\mathcal{H}_k} \tag{5}$$

$$= \left\langle f, k_{X_S} \otimes \mu_{r(X_{S^c} \mid X_S = \mathbf{x}_S)} \right\rangle_{\mathcal{H}_k}, \tag{6}$$

where $k_{X_S}$ is the product of kernels belonging to the feature set $S$, and $\mu_{r(X_{S^c} \mid X_S = \mathbf{x}_S)} := \int k_{X_{S^c}} r(X_{S^c} \mid X_S = \mathbf{x}_S) dX_{S^c}$ is the kernel mean embedding [40] of the reference distribution $r$. Depending on the choice of the reference distribution, one recovers either the standard kernel mean embedding or the conditional mean embedding. This allows us to arrive at a closed form expression of the value function and circumvents the need for fitting an exponential number of conditional densities.

## 3 Proposed method: PREF-SHAP

In this section, we will present PREF-SHAP, a new Shapley explainability toolbox designed to explain preferences by attributing contribution scores over item-level and context-level covariates for our preference models. Recall the likelihood model for C-GPM from Sec. 2.1:

$$p \left( y \mid \mathbf{u}, \mathbf{x}^{(\ell)}, \mathbf{x}^{(r)} \right) = \sigma \left( y g_U \left( \mathbf{u}, \mathbf{x}^{(\ell)}, \mathbf{x}^{(r)} \right) \right), \tag{7}$$

where $g_U$ is the context-included preference function that denotes the strength of preference of item $\mathbf{x}^{(\ell)}$ over item $\mathbf{x}^{(r)}$ under context $\mathbf{u}$. As there are two distinct sets of covariates present, we will propose two different value functions to capture the influences from items and context variables respectively, and show how they could be estimated non-parametrically using tools from the kernel methods literature, as in RKHS-SHAP.

### 3.1 Preferential value function for items

To explain a general preference model $g : \mathcal{X} \times \mathcal{X} \to \mathbb{R}$, we propose the following *preferential value function for items*.

**Definition 3.1** (Preferential value function for items). *Given a preference function $g \in \mathcal{H}$, a pair of items $(\mathbf{x}^{(\ell)}, \mathbf{x}^{(r)}) \in \mathcal{X} \times \mathcal{X}$ to compare, we define the preferential value function for items as $\nu^{(p_I)} : \mathcal{X} \times \mathcal{X} \times [0,1]^d \times \mathcal{H} \to \mathbb{R}$ such that:*

$$\nu^{(p_I)}_{\mathbf{x}^{(\ell)}, \mathbf{x}^{(r)}, S}(g) = \mathbb{E}_q \left[ g(\{X_S^{(\ell)}, X_{S^c}^{(\ell)}\}, \{X_S^{(r)}, X_{S^c}^{(r)}\}) \mid X_S^{(\ell)} = \mathbf{x}_S^{(\ell)}, X_S^{(r)} = \mathbf{x}_S^{(r)} \right] \qquad (8)$$

*where expectation is taken over the reference $q\left(X_{S^c}^{(\ell)}, X_{S^c}^{(r)} \mid X_S^{(\ell)} = \mathbf{x}_S^{(\ell)}, X_S^{(r)} = \mathbf{x}_S^{(r)}\right)$.*

We note that $\nu^{(p_I)}$ is also applicable to the context-specific preference models. For example, applying $\nu^{(p_I)}$ to $g_{\mathbf{u}} := g_U(\mathbf{u}, \cdot, \cdot)$ allows one to quantify the item covariate's influences under a specific context $\mathbf{u}$, while applying $\nu^{(p_I)}$ to $\bar{g} := \mathbb{E}_{p(U)}[g_U(U, \cdot, \cdot)]$ quantifies the average influence from each of the item covariates instead.

Similar to standard value functions, the influence of a feature set $S$ shared by the items $\mathbf{x}^{(\ell)}, \mathbf{x}^{(r)}$ is measured as the impact on the preference model after "removing" contributions from features in $S^c$, via integration with respect to some reference distribution $r$. Similar to $g$, this value function is skew-symmetric in its first two arguments, i.e. $\nu^{(p_I)}(\mathbf{x}^{(\ell)}, \mathbf{x}^{(r)}, S, g) = -\nu^{(p_I)}(\mathbf{x}^{(r)}, \mathbf{x}^{(\ell)}, S, g)$. This is justified, since features that "encourage" preference of $\mathbf{x}^{(\ell)}$ over $\mathbf{x}^{(r)}$ should naturally be the ones that "discourage" preference of $\mathbf{x}^{(r)}$ over $\mathbf{x}^{(\ell)}$ to ensure consistency. In this paper, we assume the items are i.i.d sampled from some distribution $p$, and we utilise the observational data distribution as reference as in [33], i.e. we take $r\left(X_{S^c}^{(\ell)}, X_{S^c}^{(r)} \mid X_S^{(\ell)} = \mathbf{x}_S^{(\ell)}, X_S^{(r)} = \mathbf{x}_S^{(r)}\right)$ to be $p\left(X_{S^c}^{(\ell)} \mid X_S^{(\ell)} = \mathbf{x}_S^{(\ell)}\right) p\left(X_{S^c}^{(r)} \mid X_S^{(r)} = \mathbf{x}_S^{(r)}\right)$. Although we decide here to use the observational distribution as the reference, the corresponding estimation procedure follows analogously if one instead uses the marginal distribution approach in Janzing et al. [32].

**Problems with direct application of SHAP to preference model $g$** A naive way of explaining with SHAP a general preference model $g$ which assumes no rankability would require concatenation of the items' covariates. Namely, we would set $\mathbf{z} = (\mathbf{x}^{(\ell)}, \mathbf{x}^{(r)}) \in \mathbb{R}^{2d}$ and then apply SHAP to the function $g(\mathbf{z})$ directly, now giving $2d$ Shapley values for each observed preference, i.e. two Shapley values for each feature. Not only does this approach require us to consider a larger number of feature coalitions during computation (squaring the original amount), but it also ignores that $\mathbf{x}^{(\ell)}$ and $\mathbf{x}^{(r)}$ in fact consist of the same features, leading to inconsistent explanations, i.e. that the same feature in $\mathbf{x}^{(\ell)}$ and $\mathbf{x}^{(r)}$ has a different influence, hence giving different explanations simply due to the ordering of items. We illustrate these pitfalls of such a naive approach in Appendix B.

**Empirical estimation of the preferential value function $\nu^{(p_I)}_{\mathbf{x}^{(\ell)}, \mathbf{x}^{(r)}, S}(g)$** While the *preferential value function* is general in the sense that it could be applied to any preference function $g$, we divert our attention to functions in $\mathcal{H}_{k_E}$, where $k_E$ is the *generalised preferential kernel* introduced in Sec 2.1. This allows us to adapt the recently introduced RKHS-SHAP to our settings, and we can thus circumvent learning an exponential number of conditional densities as in [33]. In the following segment, we prove the existence of the Riesz representation of the *preferential value functional*, a necessary step to adapt the RKHS-SHAP framework to our setting.

**Proposition 3.1** (Preferential value functional for items). *Let $k$ be a product kernel on $\mathcal{X}$, i.e. $k(\mathbf{x}^{(\ell)}, \mathbf{x}^{(r)}) = \prod_{j=1}^d k^{(j)}(x^{(j)}, x'^{(j)})$. Assume $k^{(j)}$ are bounded for all $j$, then the Riesz representation of the functional $\nu^{(p)}_{\mathbf{x}^{(\ell)}, \mathbf{x}^{(r)}, S}$ exists and takes the form:*

$$\nu^{(p)}_{\mathbf{x}^{(\ell)}, \mathbf{x}^{(r)}, S} = \frac{1}{\sqrt{2}} \left( \mathcal{K}(\mathbf{x}^{(\ell)}, S) \otimes \mathcal{K}(\mathbf{x}^{(r)}, S) - \mathcal{K}(\mathbf{x}^{(r)}, S) \otimes \mathcal{K}(\mathbf{x}^{(\ell)}, S) \right)$$

*where $\mathcal{K}(\mathbf{x}, S) = k_S(\cdot, \mathbf{x}_S) \otimes \mu_{X_{S^c} | X_S = \mathbf{x}_S}$ and $k_S(\cdot, \mathbf{x}_S) = \bigotimes_{j \in S} k^{(j)}(\cdot, x^{(j)})$ is the sub-product kernel defined analogously as $X_S$.*

All proofs are included in the appendix. By representing the functionals as elements in the corresponding RKHS, we can now estimate the value function non-parametrically using kernel mean embeddings.

**Proposition 3.2** (Non-parametric Estimation). *Given $\hat{g} = \sum_{j=1}^m \alpha_j k_E((\mathbf{x}_j^{(\ell)}, \mathbf{x}_j^{(r)}), \cdot)$, datasets $\mathbf{X}^{(\ell)}, \mathbf{X}^{(r)}$, test items $\mathbf{x}^{(\ell)}, \mathbf{x}^{(r)}$, the preferential value function at test items $\mathbf{x}^{(\ell)}, \mathbf{x}^{(r)}$ for coalition $S$*

Table 1: A summary of how our preference value functions can tackle different explanation tasks

| Candidate | Explanation of interest | Value function | Preference function |
|---|---|---|---|
| $\mathbf{x}^{(\ell)}, \mathbf{x}^{(r)}$ | Which item features contributed most to this duel? | $\nu^{(p_I)}_{\mathbf{x}^{(\ell)},\mathbf{x}^{(r)},S}$ | $g, \mathbb{E}_U[g_U(U,\cdot,\cdot)]$ |
| $\mathbf{x}^{(\ell)}$ | Which item features contributed most to $\mathbf{x}^{(\ell)}$'s matches? | $\frac{1}{n}\sum_{i=1}^{n}\nu^{(p_I)}_{\mathbf{x}^{(\ell)},\mathbf{x}_i,S}$ | $g, \mathbb{E}_U[g_U(U,\cdot,\cdot)]$ |
| $\mathbf{u}, \mathbf{x}^{(\ell)}, \mathbf{x}^{(r)}$ | Which context features contributed most to this duel? | $\nu^{(p_U)}_{\mathbf{u},\mathbf{x}^{(\ell)},\mathbf{x}^{(r)},S}$ | $g_U$ |
| $\mathbf{u}$ | Which context features contributed most on average? | $\frac{1}{m}\sum_{j=1}^{m}\nu^{(p_U)}_{\mathbf{u},\mathbf{x}^{(\ell)}_j,\mathbf{x}^{(r)}_j,S'}$ | $g_U$ |

*and preference function $\hat{g}$ can be estimated as*

$$\hat{\nu}^{(p_I)}_{\mathbf{x}^{(\ell)},\mathbf{x}^{(r)},S}(\hat{g}) = \boldsymbol{\alpha}^{\top}\left(\Gamma(\mathbf{X}^{(\ell)}_S,\mathbf{x}^{(\ell)}_S)\odot\Gamma(\mathbf{X}^{(r)}_S,\mathbf{x}^{(r)}_S) - \Gamma(\mathbf{X}^{(\ell)}_S,\mathbf{x}^{(r)}_S)\odot\Gamma(\mathbf{X}^{(r)}_S,\mathbf{x}^{(\ell)}_S)\right),$$

*where* $\Gamma(\mathbf{X}^{(\ell)}_S,\mathbf{x}^{(\ell)}_S) = \mathbf{K}_{\mathbf{X}^{(\ell)}_S,\mathbf{x}^{(\ell)}_S}\odot\mathbf{K}_{\mathbf{X}^{(\ell)}_{S^c},\mathbf{X}_{S^c}}\mathbf{K}^{-1}_{\mathbf{X}_S,\lambda}\mathbf{K}_{\mathbf{X}_S,\mathbf{x}^{(\ell)}_S}, \mathbf{K}_{\mathbf{X}_S,\lambda} = \mathbf{K}_{\mathbf{X}_S,\mathbf{X}_S} + n\lambda I, \boldsymbol{\alpha} = \{\alpha_j\}_{j=1}^{m}$ *and* $\lambda > 0$ *is a regularisation parameter.*

## 3.2 Preferential value function for contexts

The influence an individual context feature in $U$ has on a C-GPM function $g_U$ can be measured by the following value function.

**Proposition 3.3** (Preferential value function for contexts). *Given a preference function $g_U \in \mathcal{H}_{k_E^U}$, denote $\Omega' = \{1,...,d'\}$, then the utility of context features $S' \subseteq \Omega'$ on $\{\mathbf{u},\mathbf{x}^{(\ell)},\mathbf{x}^{(r)}\}$ is measured by $\nu^{(p_U)}_{\mathbf{u},\mathbf{x}^{(\ell)},\mathbf{x}^{(r)},S'}(g_U) = \mathbb{E}[g_U(\{\mathbf{u}_S,U_{S^c}\},\mathbf{x}^{(\ell)},\mathbf{x}^{(r)}) \mid U_S = \mathbf{u}_S]$ where the expectation is taken over the observational distribution of $U$. Now, given a test triplet $(\mathbf{u},\mathbf{x}^{(\ell)},\mathbf{x}^{(r)})$, if $\hat{g}_U = \sum_{j=1}^{m}\alpha_j k_E^U\left((\mathbf{u}_j,\mathbf{x}^{(\ell)}_j,\mathbf{x}^{(r)}_j),\cdot\right)$, the non-parametric estimator is:*

$$\hat{\nu}^{(p_U)}_{\mathbf{u},\mathbf{x}^{(\ell)},\mathbf{x}^{(r)},S'}(\hat{g}_U) = \boldsymbol{\alpha}^{\top}\left(\left(\mathbf{K}_{\mathbf{U}_{S'},\mathbf{u}_{S'}}\odot\mathbf{K}_{\mathbf{U}_{S'^c},\mathbf{U}_{S'^c}}\left(\mathbf{K}_{\mathbf{U}_{S'},\mathbf{U}_{S'}}+m\lambda'I\right)^{-1}\mathbf{K}_{\mathbf{U}_{S'},\mathbf{u}_{S'}}\right)\odot\Xi_{\mathbf{x}^{(\ell)},\mathbf{x}^{(r)}}\right)$$

*where* $\Xi_{\mathbf{x}^{(\ell)},\mathbf{x}^{(r)}} = \left(\mathbf{K}_{\mathbf{X}^{(\ell)},\mathbf{x}^{(\ell)}}\odot\mathbf{K}_{\mathbf{X}^{(r)},\mathbf{x}^{(r)}} - \mathbf{K}_{\mathbf{X}^{(r)},\mathbf{x}^{(\ell)}}\odot\mathbf{K}_{\mathbf{X}^{(\ell)},\mathbf{x}^{(r)}}\right).$

Analogously, the average influence of a specific context feature can be computed by taking an average over all pairs of matches, i.e. by using a modified value function $\frac{1}{m}\sum_{j=1}^{m}\nu^{(p_U)}_{\mathbf{u},\mathbf{x}^{(\ell)}_j,\mathbf{x}^{(r)}_j,S'}(\hat{g}_U)$. We summarise different ways to modify the proposed preferential value functions to interrogate the preference models in Table 1.

**Computational complexity of PREF-SHAP** When computing GPM, it is fundamentally a *kernel ridge regression* (KRR), which naïvely has complexity $\mathcal{O}(n^3)$. There exists a multitude of approximation techniques for KRR, the most common type being the Nyström approximation [41]. For all our experiments, we use FALKON [42], a large-scale library for solving kernel logistic regression using preconditioned conjugate gradient descent and Nyström approximations. FALKON has a computational complexity of $\mathcal{O}(n\sqrt{n})$, which effectively becomes the complexity for GPM when using FALKON. As the value function for GPM requires estimating conditional mean embeddings, which in turn also are KRR's, one can appeal to FALKON again to reduce complexity to $\mathcal{O}(n\sqrt{n})$. We summarize the procedure of PREF-SHAP in Algorithm 1. We further detail computational details pertaining to computing coalitions $S$ and batched conjugate gradient descent (`BatchedCGD`) in Appendix A.

## 4 Experiments

The main focus of our experiments is to illustrate the difference between explaining GPM (PREF-SHAP) and applying SHAP to UPM, thus highlighting the difference in explaining the mechanism of eliciting preferences and explaining the utility. When we explain UPM, we first explain how items $\mathbf{x}^{(\ell)}, \mathbf{x}^{(r)}$ affect their utilities $f(\mathbf{x}^{(\ell)}), f(\mathbf{x}^{(r)})$. Explaining the utility corresponds to calculating the value functions of the utilities $\nu_{\mathbf{x}^{(\ell)},S}(f)$ and $\nu_{\mathbf{x}^{(r)},S}(f)$. By linearity of SHAP values [14] and the simple structure relating preference and utilities in UPM, we can explain UPM by subtracting the Shapley values of $\mathbf{x}^{(\ell)}$ with $\mathbf{x}^{(r)}$. However, this type of explanation is only correct when data is rankable, which seldom happens in practice, thus motivating PREF-SHAP.

**Algorithm 1** PREF-SHAP

---

**Input:** Solution $\boldsymbol{\alpha}$, datasets $\mathbf{X}^{(\ell)}, \mathbf{X}^{(r)}, \mathbf{X}, \mathbf{U}$, test items $\mathbf{x}^{(\ell)}, \mathbf{x}^{(r)}, \mathbf{u}$, batch size $n_b$, number of coalition samples $n_S$, context-specific flag `cflg`
1: Compute effective dimension $d_{\text{eff}} :=$ Number of features with variance greater than 0.
2: Compute coalitions $\mathcal{S} = \{S_1, \ldots, S_{n_S}\}$, form binary matrix $\mathbf{Z} \in \{0,1\}^{n_S, d_{\text{eff}}}$ from $\mathcal{S}$, and compute weights $\mathbf{W} = [w_1, \ldots, w_{n_S}]$ with $w_i = \frac{d-1}{\binom{d}{|S_i|}|S_i|(d-|S_i|)}$.
3: **for** batch $\mathcal{S}_b$ in $\mathcal{S}$ **do**
4:    **if** `cflg` Take $S, S^c$ of $\mathbf{X}^{(\ell)}, \mathbf{X}^{(r)}, \mathbf{X}, \mathbf{x}^{(\ell)}, \mathbf{x}^{(r)}$ **else** Take $S', S'^c$ of $\mathbf{U}, \mathbf{u}$
5:    **if** `cflg` Compute $\mathbf{K}_{\mathbf{X}_S, \lambda}^{-1}[\mathbf{K}_{\mathbf{X}_S, \mathbf{x}_S^{(\ell)}}, \mathbf{K}_{\mathbf{X}_S, \mathbf{x}_S^{(r)}}]$ **else** $\left(\mathbf{K}_{\mathbf{U}_{S'}, \mathbf{U}_{S'}} + m\lambda' I\right)^{-1} \mathbf{K}_{\mathbf{U}_{S'}, \mathbf{u}_{S'}}$ using `BatchedCGD`
6:    **if** `cflg` Compute $\hat{\nu}_{\mathbf{x}^{(\ell)}, \mathbf{x}^{(r)}, \mathcal{S}_b}^{(p)}(\hat{g})$ **else** $\hat{\nu}_{\mathbf{u}, \mathbf{x}^{(\ell)}, \mathbf{x}^{(r)}, \mathcal{S}_b'}(\hat{g}_U)$
7: **end for**
8: **if** `cflg` Set $\mathbf{v}_x = \{\hat{\nu}_{\mathbf{x}^{(\ell)}, \mathbf{x}^{(r)}, \mathcal{S}_b}^{(p)}(\hat{g})\}_{b=1}^B$ **else** $\mathbf{v}_x = \{\hat{\nu}_{\mathbf{u}, \mathbf{x}^{(\ell)}, \mathbf{x}^{(r)}, \mathcal{S}_b'}(\hat{g}_U)\}_{b=1}^B$
9: Calculate Shapley values $\boldsymbol{\beta}_x = \left(\mathbf{Z}^\top \mathbf{W} \mathbf{Z}\right)^{-1} \mathbf{Z}^\top \mathbf{W} \mathbf{v}_x$
10: **return** $\boldsymbol{\beta}_x$

---

We apply PREF-SHAP to unrankable synthetic and real-world datasets to connect theory with practice. We split data, i.e. matches with their outcomes, into train (80%), validation (10%), and test (10%) and explain the model on a random subset of the data. The hyperparameters for the kernels are selected using gradient descent, based on the proposed method in [43]. We first generate synthetic duelling data where performance can be compared against ground truth, to demonstrate that PREF-SHAP is capable of identifying the relevant features.

**Synthetic data** We first consider a synthetic experiment with unrankable duelling data. We generate the items by first sampling 1000 item covariates $[x_i^{[0]}, x_i^{AB}, x_i^{AC}, x_i^{BC}] =: \mathbf{p}_i \in \mathbb{R}^4 \sim \mathcal{N}(0, \mathbf{I}_4)$. We associate each item with a cluster membership $c_i \in \{A, B, C\}$, where the assignment is randomly chosen for each item with equal probability. We then form the full item covariate by concatenating $\mathbf{p}_i$ with one-hot encoded $c_i$ as $\mathbf{x}_i = [\mathbf{p}_i, \texttt{one\_hot}(c_i)]$. 40000 matches between randomly chosen pairs of items are conducted by the following mechanism: match outcomes are decided based on the underlying cluster membership of the items. For example, if an item from cluster $A$ competes against an item from cluster $B$, the winner is decided by their inter-cluster covariate $x^{AB}$, i.e. $i \preceq j$ if $x_j^{AB} \geq x_i^{AB}$. When the match is between members of the same cluster, it is dictated by the maximum among the within-cluster variable, i.e. $\max(x_i^{[0]}, x_j^{[0]})$. See Fig. 1 for an illustration. As no clusters have any advantage over the others, the data is not rankable, and we expect the inter-cluster covariates $x^{AB}, x^{AC}, x^{BC}$ to have similar explanations on average, but significantly different from each other when we examine local explanations.

We consider both global and *grouped-local* explanations of the synthetic dataset in Figure 2 and Figure 3 respectively. In the global explanations, we explain all matches regardless of the cluster membership, while in the grouped-local explanations we only explain matches between items from $A$ against items from $B$. For more grouped-local explanations on different cluster pairs, we refer to appendix B.

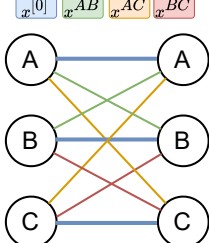

**Interpreting the simulation explanations.** The beehive plots showcase the recovered PREF-SHAP values, where the bar plots demonstrated the average PREF-SHAP values for each feature. The colour in the beehive plots indicates the magnitude of the difference between the corresponding features of the winner and of the loser in that match. For example, a red point in a beehive plot for feature $d$ indicates that the difference $x_{winner}^{(d)} - x_{loser}^{(d)}$ is large.

Figure 1: An illustration of our simulation: each edge corresponds to the variable that dictates the comparison based on the colour.

Fig. 2 illustrates the explanation results for the global synthetic experiments. We see that PREF-SHAP identified the within-cluster variable $x^{[0]}$ as the most important, which is a consequence of the fact that the largest number of matches are played between the items of the same cluster (cf. Fig. 1 where there are three blue lines and two lines of each of the other colours). The three inter-cluster

Table 2: Dataset summary

| Dataset | $N_{\text{Matches}}$ | $N_{\text{items}}$ | $N_{\text{Context}}$ | $D_{\text{continuous}}$ | $D_{\text{binary}}$ | $D_{\text{continuous}}^{\text{Context}}$ | $D_{\text{binary}}^{\text{Context}}$ |
|---|---|---|---|---|---|---|---|
| *Synthetic* | 40000 | 1000 | - | 4 | 3 | - | - |
| *Chameleon* | 106 | 35 | - | 7 | 19 | - | - |
| *Pokémon* | 60000 | 800 | - | 7 | 0 | - | - |
| *Tennis* | 95359 | 3483 | 4114 (tournaments) | 4 | 7 | 0 | 6 |

variables contributed similarly according to PREF-SHAP, which, by symmetry, should be the case. Furthermore, the correct battle mechanism is captured by PREF-SHAP but not UPM, as we see that the large PREF-SHAP values for each feature are red in the beehive plot. This indicates that items with larger value are more likely to win against items with lower value in the corresponding features. In contrast, SHAP for UPM does not recover this insight.

The explanations for the matches between items from $A$ against items from $B$, are shown in Fig 3. Here, $x^{AB}$ is correctly picked as the relevant feature in these matches with PREF-SHAP, but not with SHAP for UPM. We see again that there is a clear tendency that large PREF-SHAP values are red for feature $x^{AB}$, showing that PREF-SHAP once again captures the designed gaming mechanism – which is not the case in SHAP for UPM. Intuitively, even though SHAP for UPM allows local explanations, it does so based on a *global utility*, which fails completely in a non-rankable case.

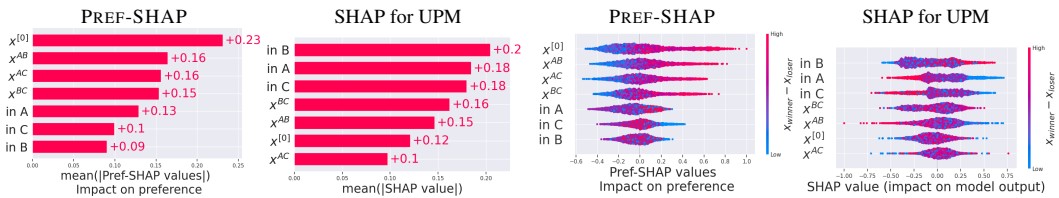

Figure 2: Bar and Beehive plots for global explanations on the synthetic dataset.

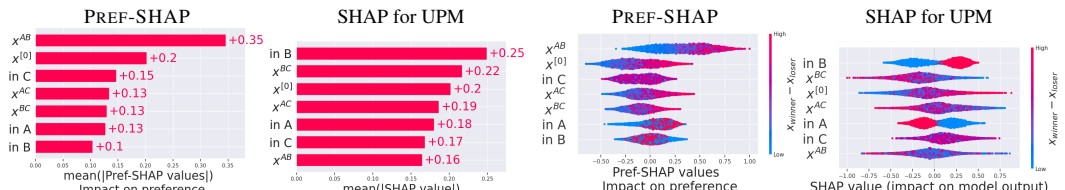

Figure 3: Bar and Beehive plots for grouped local explanations on the synthetic dataset (Cluster $A$ vs $B$).

**Real-world explanations** For our real-world datasets, we consider publicly available datasets *Chameleon*, *Pokémon* and *Tennis*. We provide descriptive statistics of these datasets in Table 5 and give their brief descriptions below. Appendix B contains further large scale experiments on an additional dataset consisting of user-item interactions on a fashion retail website.

*The Chameleon* dataset [44] considers 106 contests between 35 male dwarf chameleons. Physical traits of the chameleons are measured such as the *height of their casque*, *length of their jaw*, *body mass* etc. According to [44], they fitted a linear Bradley Terry model and examined the coefficients to deduce that *casque height* (ch.res) and *relative area of the flank patch* (prop.patch) positively affected the fighting ability the most. *The Pokémon* dataset considers 60000 Pokémon battles among 800 Pokémon. Pokémon have different characteristics such as *attack power*, *speed*, *health* etc. The Pokémon further has at least one different *type* such as *Electric, Water, Fire*, etc. Certain types have advantages and disadvantages against each other, for instance, fire Pokémon are weak to water-based attacks (receiving twice the damage) and as a result have a disadvantage against water Pokémon.

*The Tennis* dataset considers professional tennis matches between 1991 and 2017 in all major tournaments each year. The data is provided publicly by ATP World Tour [45]. Features such as *birthyear*, *weight*, *height* etc are included about each tennis player together with context details of the match such as the court being indoor or outdoor and what surface the match is being played on.

The above datasets are not rankable, and we validate this claim by comparing GPM performance against UPM in Table 4, together with the estimated rankability measure *SpecR* proposed in [23] for each dataset. *SpecR* measures the similarity of the data to a complete dominance graph (i.e. rankable data). It takes values between 0 and 1 with values close to 1 being evidence in support of rankability.

Table 3: GPM vs UPM. Mean and standard deviations of performance averaged over 5 runs.

| | Synthetic | | Chameleon | | Pokémon | | Tennis | |
|---|---|---|---|---|---|---|---|---|
| | GPM | UPM | GPM | UPM | GPM | UPM | C-GPM | UPM |
| Test AUC | $0.98_{\pm 0.00}$ | $0.71_{\pm 0.01}$ | $0.92_{\pm 0.07}$ | $0.80_{\pm 0.07}$ | $0.86_{\pm 0.00}$ | $0.82_{\pm 0.00}$ | $0.58_{\pm 0.02}$ | $0.52_{\pm 0.02}$ |
| SpecR | 0.09 | | 0.24 | | 0.20 | | $0.13_{\pm 0.07}$ | |

For the Tennis data where there are additional relationships with the context (tournaments), we estimate the average *SpecR* of each tournament. Both the superior performance of GPM over UPM and the low *SpecR* measures suggest that the datasets are generally not rankable, which points to limitations of explaining preferences via utility-based modeling.

**Explaining Pokémon battles.** We first consider standard dueling data for explaining preferences. We explain the learned preferences and learned differences in utilities on the Pokémon dataset in Figure 4. In this dataset, we have summed the Shapley values for *Type* features.

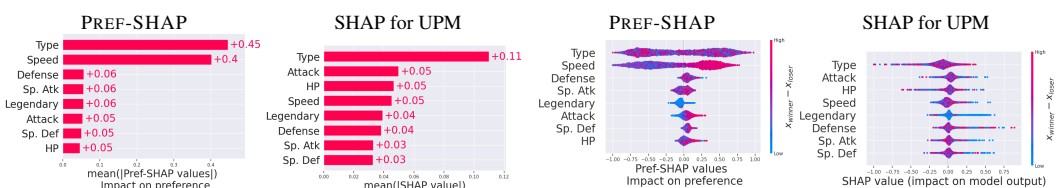

Figure 4: Bar and Beehive plots for the Pokémon dataset. PREF-SHAP captures that both speed and type matter, while SHAP for UPM only captures the type importance.

We see that explaining general preferences provides further insight than just explaining the difference in utility functions. In particular, SHAP for UPM does not capture the additional importance of *Speed* in winning battles. As higher (more red) values of differences in speed $x^{\text{speed}}_{\text{winner}} - x^{\text{speed}}_{\text{loser}}$ have positive impact on the outcome, we conclude that having higher speed than your opponent is advantageous besides a type advantage. This insight is aligned with the "Sweeper" strategy [46], where one would employ a leading Pokémon with very high speed and attack to attempt downing the opponent before they can strike back. In Fig. 5, we see PREF-SHAP can also capture the correct type advantage/disadvantages among the Pokémon, but not SHAP for UPM.

**Explaining Chameleon contests.** We find that UPM's explanations are more aligned with [44]'s findings (*prop.path* and *ch.res* are the most important features), which is unsurprising since the Bradley Terry model used in [44] is also a utility based model. However, since GPM gives a much better predictive performance than UPM (Test AUC 0.92 v.s. 0.80), we believe PREF-SHAP's explanations are also insightful. In fact, PREF-SHAP discovers that having larger *jaw sizes* (jl.res) than your opponent have a significant negative effect on match outcome, a previously undiscovered mechanism from [44]. We verify this finding in Appendix B by applying PREF-SHAP to GPM trained on multiple folds of the Chameleon dataset and consistently find that high values of the *jaw size* (jl.res) variable have a negative impact on the outcome.

Figure 5: Explaining matches between 4 types of Pokémon, among them only fire and water has a type disadvantage/advantage against each other. PREF-SHAP (top) correctly identifies that fire and water are the most important, while water and fire are not deemed most important by SHAP for UPM.

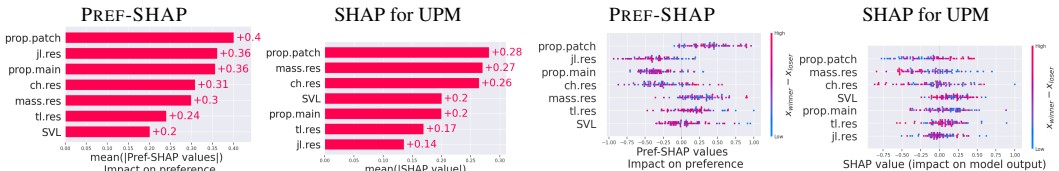

Figure 6: Bar and Beehive plots for the Chameleon dataset

**Explaining Tennis matches.** We now consider preference learning with context covariates and explain both item characteristics and context covariates in Figure 7. In terms of item-based inference, PREF-SHAP finds that being older than your opponent ($x^{\text{yob}}_{\text{winner}} - x^{\text{yob}}_{\text{loser}} < 0 \rightarrow$ Blue), physically

heavier, and taller than your opponent positively impacts the chances of winning. We also find that debuting earlier as a professional tennis player than your opponent positively impacts your chances of winning. This is not surprising as debuting earlier may be indicative of a promising young talent. Across all competitions, there appear to be no significant patterns in environment effects.

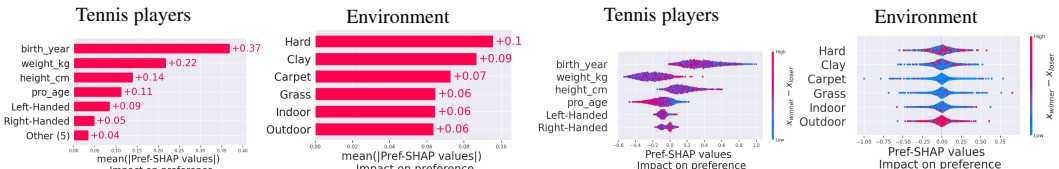

Figure 7: Item and context-specific Pref-SHAP values for the Tennis dataset

**Explaining Djokovic's losses** In plot Figure 8, we locally explain all Novak Djokovic's losses in his professional career. Novak Djokovic is regarded as one of the greatest tennis players of all time, so understanding his weakness could serve as a practical demonstration of the utility of PREF-SHAP.

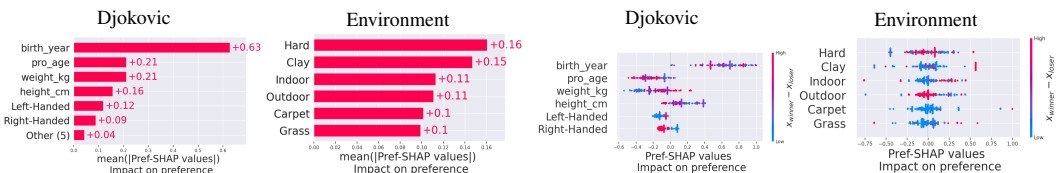

Figure 8: Local explanations of Djokovic losses

While the results take a similar shape to the global explanations, Djokovic remarkably seems to be weaker to players shorter than him, contrary to the general advantage of being taller. Besides this, Djokovic seems to be weaker on clay courts and when playing indoors.

# 5   Conclusion

In this work, we proposed PREF-SHAP to explain preference learning for pairwise comparison data. We proposed the appropriate value function for preference explanations and demonstrated the pathologies of the naive concatenation approach in Appendix B. Experiments demonstrated that PREF-SHAP recovers richer explanations than utility-based approaches, showcasing the ability of PREF-SHAP in interpreting the mechanism of preference elicitation.

# Acknowledgments

The authors sincerely thank Oscar Clivio for the helpful comments and feedback. SLC is supported by the EPSRC and MRC through the OxWaSP CDT programme EP/L016710/1. DS is supported in part by Tencent AI Lab and in part by the Alan Turing Institute (EP/N510129/1).

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
