# A    Computation and Implementation Details

We propose several optimizations in the PREF-SHAP procedure. We consider fast sampling of coalitions $S$ in Algorithm 2 batched conjugate gradient descent in Algorithm 3 described below.

**Fast coalitions**    We first propose an optimized sampling scheme for finding coalitions $S$ in Algorithm 2. In contrast to the implementation in [14] which samples the weights from $p(Z)$, our method

---

**Algorithm 2** Sampling unique $n$, $d$-dimensional coalitions in $\mathcal{O}(d)$ time

---

**Input:** Number of coalitions $n$, number of features $d$.

1: $I = \text{SampleWithoutReplacement}(n, 0, 2^d)$      $\triangleright$ Sample $n$ unique integers between 0 and $2^d$
2: **def** base2(i):      $\triangleright$ Convert integer to base-2 representation
3:     $S = [0, \ldots, 0] \in \mathbb{R}^d, r = i$      $\triangleright$ Initialize $d$-dimensional 0 vector and the rest term $r$
4:     **while** $r > 0$ **do**
5:        $i = \lfloor \log_2(r) \rfloor$      $\triangleright$ Find which index of $S$ to set to 1
6:        $S[i] = 1$      $\triangleright$ Update $S$
7:        $r = r - 2^i$      $\triangleright$ Update rest term
8:     **end while**
9:     **return** $S$
    **return** $\{S_1, \ldots S_n\} = \text{parallel\_apply}(I, \text{base2})$ $\triangleright$ Each integer can be independently converted

---

is embarrassingly parallel, which allows for an additional $\mathcal{O}(n)$ reduction. A naive algorithm that compares each sample $S_i$ has complexity $\mathcal{O}(n^2 d^2)$ and cannot be parallelized.

**Stabilizing the Shapley value estimation**    We remove the features which have 0 variance in the data we are explaining, similar to the implementation in SHAP. To ensure we get numerically stable Shapley Values, we calculate the inverse using Cholesky decomposition, as we found the regular inverse function provided inconsistent results.

To calculate CMEs effectively, we use *preconditioned batched* conjugate gradient descent over coalitions detailed in Algorithm 3.

---

**Algorithm 3** Batched conjugate gradient descent

---

**Input:** Preconditioner $P = \mathbf{K}_{\mathbf{x}_D}^{-1}$, batch $\mathbf{X} = [\mathbf{K}_{\mathbf{x}_{S_i}} \ldots \mathbf{K}_{\mathbf{x}_{S_n}}]$, $\mathbf{B} = [\mathbf{K}_{\mathbf{x}_{S_i}, x_{S_i}} \ldots \mathbf{K}_{\mathbf{x}_{S_n}, x_{S_n}}]$,
   max_its, tolerance $\varepsilon$
   Set $\mathbf{R} = \mathbf{B}, \mathbf{Z} = \text{BatchMM}(P, \mathbf{B}), \mathbf{p} = \mathbf{Z}, \mathbf{a} = \mathbf{0}$
   Set $\mathbf{R}_Z = [(\mathbf{R}_i \circ \mathbf{Z}_i)_{++} \ldots (\mathbf{R}_n \circ \mathbf{Z}_n)_{++}]$      $\triangleright$ Element wise product and sum
   **for** max_its **do**
      $\mathbf{L} = \text{BatchMM}(\mathbf{X}, \mathbf{B})$
      $\boldsymbol{\alpha} = \mathbf{L} \circ \frac{1}{[(\mathbf{L}_i \circ \mathbf{p}_i)_{++} \ldots (\mathbf{L}_n \circ \mathbf{p}_n)_{++}]}$
      $\mathbf{a} = \mathbf{a} + \boldsymbol{\alpha} \circ \mathbf{p}$
      $\mathbf{r} = \mathbf{r} - \boldsymbol{\alpha} \circ \mathbf{L}$
      **if** $\text{Mean}([(\mathbf{r}_i \circ \mathbf{r}_i)_{++} \ldots (\mathbf{r}_n \circ \mathbf{r}_n)_{++}]) < \varepsilon$ **then return** $\mathbf{a}$
      **end if**
      $\mathbf{z} = \text{BatchMM}(P, \mathbf{r})$
      $\mathbf{R}_Z^{\text{new}} = [(\mathbf{R}_i \circ \mathbf{Z}_i)_{++} \ldots (\mathbf{R}_n \circ \mathbf{Z}_n)_{++}]$
      $\mathbf{p} = \mathbf{z} + [(\mathbf{R}_i^{\text{new}} \circ \frac{1}{\mathbf{R}_i})_{++} \ldots (\mathbf{R}_n^{\text{new}} \circ \frac{1}{\mathbf{R}_n})_{++}] \circ \mathbf{p}$
      $\mathbf{R}_Z = \mathbf{R}_Z^{\text{new}}$
   **end for**
   **return** $\mathbf{a}$

---

We have run all our jobs on one Nvidia V100 GPU.

# B    Additional Experimental Results

**Naive Concatenation**    We demonstrate the pathologies of the naive concatenation approach mentioned in Sec. 3 with our synthetic experiment. Recall that naive-concatenation approach here corresponds to first concatenating $\mathbf{x}^{(\ell)}, \mathbf{x}^{(r)}$'s features together and applying SHAP to the learned

function $g$ directly, in order to to obtain $2d$ Shapley values, instead of the original $d$, since each feature has been duplicated. This approach ignores that the items $\mathbf{x}^{(\ell)}$ and $\mathbf{x}^{(r)}$ in fact consist of the same features. Therefore, when we use the usual value function from SHAP (corresponding to the impact an individual feature has on the model when it is turned "off" by integration), we would be turning "off" the feature from the left item, while keeping "on" the feature from the right item, obtaining a difficult to interpret attribution score. This is highly problematic, as we might be inferring vastly different contributions of the same feature purely because of the item ordering when concatenating them. We note that the item ordering in all our experiments is arbitrary and carries no additional information about the match.

We can see from Fig. 9 that when we explain the preference model applied to the synthetic experiment, we see that, for example, $x^{AC}(l)$ from $\mathbf{x}^{(\ell)}$ and $x^{AC}(r)$ from $\mathbf{x}^{(r)}$ have in fact very different average Shapley values. Even attempting to average each pair of corresponding features does not give a meaningful feature contribution ordering ($x^{AB}$ and $x^{AC}$ are scored higher on average than $x^{BC}$ and $x^{[0]}$).

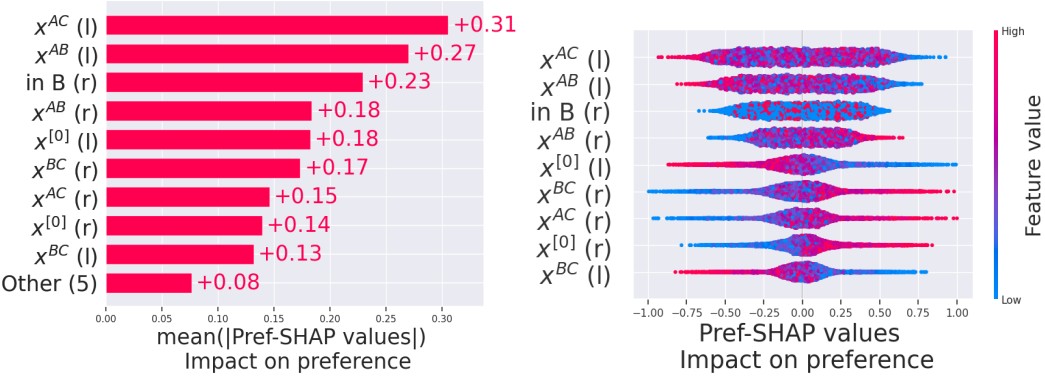

Figure 9: Explaining a naïve concatenation model

**Additional synthetic data**   We consider an additional synthetic experiment where we generate data directly from a GPM model and one where we construct synthetic dueling data. When simulating data, we first generate player covariates as $\mathbf{x}_i \in \mathbb{R}^d \sim \mathcal{N}(0, 0.1\mathbf{I}_d)$ for each player $i$. When generating from the GPM model, we would set 2 covariates as important, by only keeping the 2 first entries of $\mathbf{x}_i$ and fixing the rest to be constant (equal to 0). We build a GPM model for $g$ out of these covariates and generate match outcomes.

We consider $d = 10$, where only the two first features are set to be important in predicting the outcome.

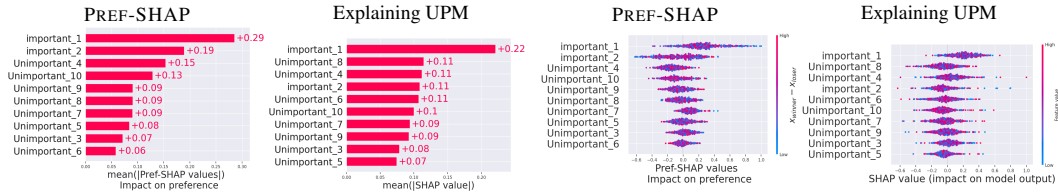

Figure 10: Bar and Beehive plots for Simulation A. PREF-SHAP recovers the correct features (1,2), while explaining UPM does not.

**Chameleon data**   We further provide explanations of the Chameleon dataset on several different folds in Figure 11 and Figure 12.

**Additional local explanations**   We additionally provide local explanations on the synthetic dataset in Figure 13 and Figure 14.

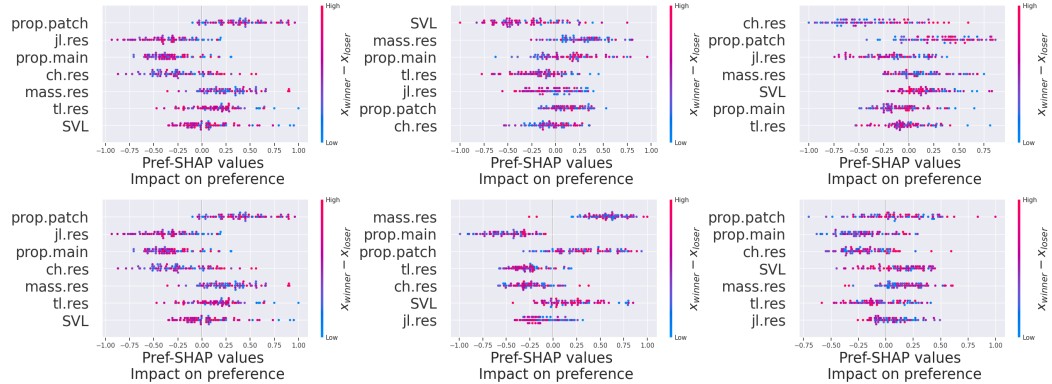

Figure 11: PREF-SHAP applied to 6 different folds of Chameleon. PREF-SHAP consistently finds that higher values of *jaw length* (jl.res) have a negative impact on outcome.

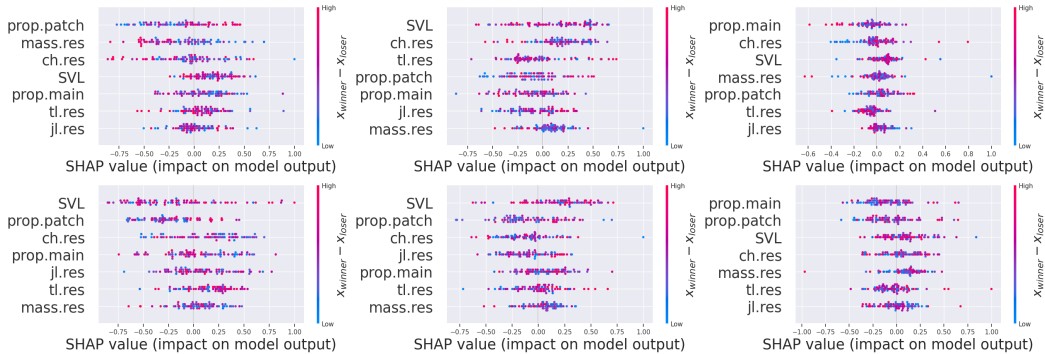

Figure 12: UPM explanations on 6 different folds of Chameleon. UPM is unable to find a consistent pattern for the impact of *jaw length* (jl.res) on the outcome.

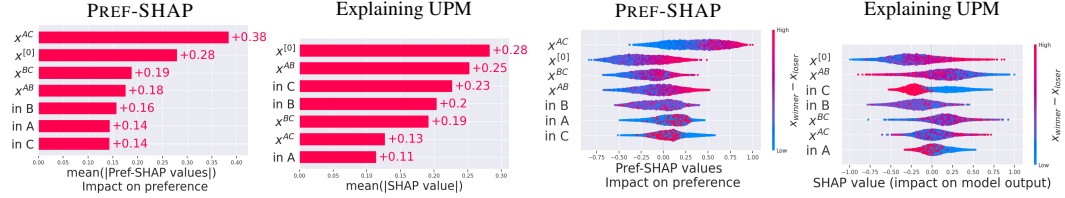

Figure 13: Explaining matches between clusters 0 and 2 on the synthetic dataset.

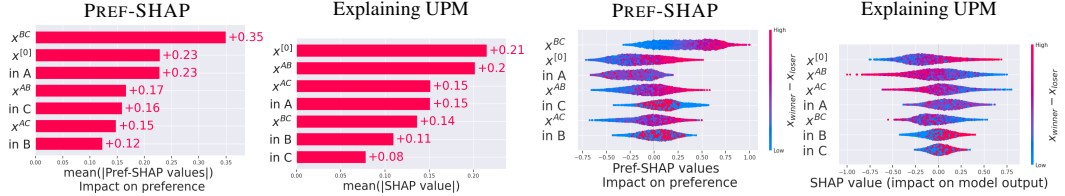

Figure 14: Explaining matches between clusters 1 and 2 on the synthetic dataset.

**Website dataset**  *The Website* dataset considers anonymized visitors on a fashion retail website, where we are given what garment each visitor viewed and what each visitor clicked in a session. A user may have more than one session. In this setup, we interpret a browsing session for a visitor as multiple matches between items, such that the winning item (clicked) competes against all losing items (only viewed). If several items are winners, they do not play against each other. Each item has several descriptive statistics such as *colour*, *garment type*, *assortment characteristic* etc. There

Table 5: Dataset summary

| Dataset | $N_{\text{Matches}}$ | $N_{\text{Players}}$ | $N_{\text{Context}}$ | $D_{\text{continuous}}$ | $D_{\text{binary}}$ | $D_{\text{continuous}}^{\text{Context}}$ | $D_{\text{binary}}^{\text{Context}}$ |
|---------|---------|---------|---------|---------|---------|---------|---------|
| *Website* | 85144 | 20626 | 129117 (users) | 0 | 93 | 1 | 4 |

are some limited descriptive statistics of the visitors, such as *year of birth* and *gender code* (i.e. Male/Female/Unspecified/Unknown).

**Explaining Website**    For the website dataset, we explain product and user preferences in Figure 15. We generally found that, for the period considered, cosmetic products and the "Jersey Basic category" drove clicks.

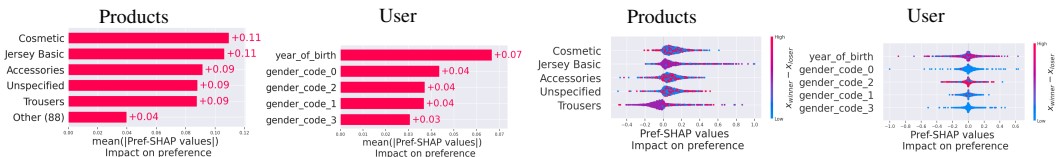

Figure 15: Barplots and beeplots for the website dataset, products on the left and user variables on the right.

Table 4: GPM vs UPM. Mean and standard deviations of performance averaged over 5 runs.

| | Synthetic | | Chameleon | | Pokémon | | Tennis | | Website | |
|---|---|---|---|---|---|---|---|---|---|---|
| | GPM | UPM | GPM | UPM | GPM | UPM | C-GPM | UPM | C-GPM | UPM |
| Test AUC | $0.98_{\pm 0.00}$ | $0.71_{\pm 0.01}$ | $0.92_{\pm 0.07}$ | $0.80_{\pm 0.07}$ | $0.86_{\pm 0.00}$ | $0.82_{\pm 0.00}$ | $0.58_{\pm 0.02}$ | $0.52_{\pm 0.02}$ | $0.66_{\pm 0.01}$ | $0.65_{\pm 0.01}$ |
| SpecR | 0.09 | | 0.24 | | 0.20 | | $0.13_{\pm 0.07}$ | | $0.53_{\pm 0.10}$ | |

# C   Proofs

**Proposition 3.1** (Preferential value functional for items). *Let $k$ be a product kernel on $\mathcal{X}$, i.e. $k(\mathbf{x}^{(\ell)}, \mathbf{x}^{(r)}) = \prod_{j=1}^{d} k^{(j)}(x^{(j)}, x'^{(j)})$. Assume $k^{(j)}$ are bounded for all $j$, then the Riesz representation of the functional $\nu_{\mathbf{x}^{(\ell)}, \mathbf{x}^{(r)}, S}^{(p)}$ exists and takes the form:*

$$\nu_{\mathbf{x}^{(\ell)}, \mathbf{x}^{(r)}, S}^{(p)} = \frac{1}{\sqrt{2}} \left( \mathcal{K}(\mathbf{x}^{(\ell)}, S) \otimes \mathcal{K}(\mathbf{x}^{(r)}, S) - \mathcal{K}(\mathbf{x}^{(r)}, S) \otimes \mathcal{K}(\mathbf{x}^{(\ell)}, S) \right)$$

*where $\mathcal{K}(\mathbf{x}, S) = k_S(\cdot, \mathbf{x}_S) \otimes \mu_{X_{S^c}|X_S = \mathbf{x}_S}$ and $k_S(\cdot, \mathbf{x}_S) = \bigotimes_{j \in S} k^{(j)}(\cdot, x^{(j)})$ is the sub-product kernel defined analogously as $X_S$.*

*Proof.*  From [16], we know the generalised preferential kernel has the following feature map:

$$k_E((\mathbf{x}^{(\ell)}, \mathbf{x}^{(r)}), \cdot) = \frac{1}{\sqrt{2}} \left( k(\cdot, \mathbf{x}^{(\ell)}) \otimes k(\cdot, \mathbf{x}^{(r)}) - k(\cdot, \mathbf{x}^{(r)}) \otimes k(\cdot, \mathbf{x}^{(\ell)}) \right) \tag{9}$$

where $\otimes$ are the usual tensor product. Recall we defined the preferential value function for items as,

$$\nu_{\mathbf{x}^{(\ell)}, \mathbf{x}^{(r)}, S}^{(p_I)}(g) = \mathbb{E}[g(X^{(\ell)}, X^{(r)}) \mid X_S^{(\ell)} = \mathbf{x}_S^{(\ell)}, X_S^{(r)} = \mathbf{x}_S^{(r)}] \tag{10}$$

as $\nu_{\mathbf{x}^{(\ell)}, \mathbf{x}^{(r)}, S}^{(p_I)}$ is a bounded linear functional on $g$ where $g \in \mathcal{H}_{k_E}$ is bounded, Riesz representation theorem [25] tells us there exists a Riesz representation of the functional in $\mathcal{H}_{k_E}$, which for notation simplicity, we will denote it as $\nu_{\mathbf{x}^{(\ell)}, \mathbf{x}^{(r)}, S}^{(p_I)}$ as well. This corresponds to,

$$\nu_{\mathbf{x}^{(\ell)}, \mathbf{x}^{(r)}, S}^{(p_I)}(g) = \mathbb{E}[g(X^{(\ell)}, X^{(r)}) \mid X_S^{(\ell)} = \mathbf{x}_S^{(\ell)}, X_S^{(r)} = \mathbf{x}_S^{(r)}] \tag{11}$$

$$= \langle g, \mathbb{E}[k_E\left((X^{(\ell)}, X^{(r)}), \cdot\right) \mid X_S^{(\ell)} = \mathbf{x}_S^{(\ell)}, X_S^{(r)} = \mathbf{x}_S^{(r)}] \rangle_{\mathcal{H}_{k_E}} \tag{12}$$

$$= \langle g, \nu_{\mathbf{x}^{(\ell)}, \mathbf{x}^{(r)}, S}^{(p_I)} \rangle_{\mathcal{H}_{k_E}} \tag{13}$$

now we expand the expectation of the feature map as,

$$\nu^{(p_I)}_{\mathbf{x}^{(\ell)},\mathbf{x}^{(r)},S} = \mathbb{E}\left[\frac{1}{\sqrt{2}}\left(k(\cdot,X^{(\ell)}) \otimes k(\cdot,X^{(r)}) - k(\cdot,X^{(r)}) \otimes k(\cdot,X^{(\ell)})\right) \mid X_S^{(\ell)} = \mathbf{x}_S^{(\ell)}, X_S^{(r)} = \mathbf{x}_S^{(r)}\right] \tag{14}$$

However, we note that

$$\mathbb{E}[k(\cdot,X^{(\ell)}) \otimes k(\cdot,X^{(r)}) \mid X_S^{(\ell)} = \mathbf{x}_S^{(\ell)}, X_S^{(r)} = \mathbf{x}_S^{(r)}] = \mathbb{E}[k(\cdot,X) \mid X_S = \mathbf{x}_S^{(\ell)}]$$
$$\otimes \mathbb{E}[k(\cdot,X) \mid X_S = \mathbf{x}_S^{(r)}],$$

because $X^{(\ell)}$ and $X^{(r)}$ are identical copies of $X$ and we take the reference distribution as $p(X^{(\ell)}, X^{(r)} \mid X_S^{(\ell)} = \mathbf{x}_S^{(\ell)}, X_S^{(r)} = \mathbf{x}_S^{(r)}) = p(X^{(\ell)} \mid X_S^{(\ell)} = \mathbf{x}_S^{(\ell)})p(X^{(r)} \mid X_S^{(r)} = \mathbf{x}_S^{(r)})$. Focusing on the duplicating component, we have,

$$\mathbb{E}[k(\cdot,X) \mid X_S = \mathbf{x}^{(\ell)}] = \mathbb{E}[k_S(\cdot,X_S) \otimes k_{S^c}(\cdot,X_{S^c}) \mid X_S = \mathbf{x}_S^{(\ell)}] \tag{15}$$

$$= k_S(\cdot,\mathbf{x}_S^{(\ell)}) \otimes \mathbb{E}[k_{S^c}(\cdot,X_{S^c}) \mid X_S = \mathbf{x}_S^{(\ell)}] \tag{16}$$

$$= k_S(\cdot,\mathbf{x}_S^{(\ell)}) \otimes \mu_{X_{S^c}|X_S=\mathbf{x}_S^{(\ell)}} \tag{17}$$

$$=: \mathcal{K}(\mathbf{x}^{(\ell)}, S) \tag{18}$$

therefore by symmetry, we can arrange the terms in Eq 14 and conclude the proposition,

$$\nu^{(p_I)}_{\mathbf{x}^{(\ell)},\mathbf{x}^{(r)},S} = \frac{1}{\sqrt{2}}\left(\mathcal{K}(\mathbf{x}^{(\ell)}, S) \otimes \mathcal{K}(\mathbf{x}^{(r)}, S) - \mathcal{K}(\mathbf{x}^{(r)}, S) \otimes \mathcal{K}(\mathbf{x}^{(\ell)}, S)\right) \tag{19}$$

$$\square$$

To estimate the preferential value functional, we simply replace the conditional mean embeddings with the empirical versions, i.e. $\hat{\mathcal{K}}(\mathbf{x}, S) = k_S(\cdot,\mathbf{x}_S) \otimes \hat{\mu}_{X_{S^c}|X_S=\mathbf{x}_S}$, where $\hat{\mu}_{X_{S^c}|X_S=\mathbf{x}_S} = \mathbf{K}_{\mathbf{x}_S,\mathbf{x}_S}(\mathbf{K}_{\mathbf{x}_S,\mathbf{x}_S} + n\lambda I)^{-1}\mathbf{\Phi}_{X_{S^c}}^{\top}$ is the standard conditional mean embedding estimator ($\mathbf{\Phi}_{X_{S^c}}$ is the feature map matrix of rv $X_{S^c}$).

Now we proceed to estimate the preferential value function given a function $g$ from the RKHS,

**Proposition 3.2** (Non-parametric Estimation). *Given $\hat{g} = \sum_{j=1}^{m} \alpha_j k_E((\mathbf{x}_j^{(\ell)}, \mathbf{x}_j^{(r)}), \cdot)$, datasets $\mathbf{X}^{(\ell)}, \mathbf{X}^{(r)}$, test items $\mathbf{x}^{(\ell)}, \mathbf{x}^{(r)}$, the preferential value function at test items $\mathbf{x}^{(\ell)}, \mathbf{x}^{(r)}$ for coalition $S$ and preference function $\hat{g}$ can be estimated as*

$$\hat{\nu}^{(p_I)}_{\mathbf{x}^{(\ell)},\mathbf{x}^{(r)},S}(\hat{g}) = \boldsymbol{\alpha}^{\top}\left(\Gamma(\mathbf{X}_S^{(\ell)},\mathbf{x}_S^{(\ell)}) \odot \Gamma(\mathbf{X}_S^{(r)},\mathbf{x}_S^{(r)}) - \Gamma(\mathbf{X}_S^{(\ell)},\mathbf{x}_S^{(r)}) \odot \Gamma(\mathbf{X}_S^{(r)},\mathbf{x}_S^{(\ell)})\right),$$

*where $\Gamma(\mathbf{X}_S^{(\ell)},\mathbf{x}_S^{(\ell)}) = \mathbf{K}_{\mathbf{X}_S^{(\ell)},\mathbf{x}_S^{(\ell)}} \odot \mathbf{K}_{\mathbf{X}_{S^c}^{(\ell)},\mathbf{X}_{S^c}}\mathbf{K}_{\mathbf{X}_S,\lambda}^{-1}\mathbf{K}_{\mathbf{X}_S,\mathbf{x}_S^{(\ell)}}, \mathbf{K}_{\mathbf{X}_S,\lambda} = \mathbf{K}_{\mathbf{X}_S,\mathbf{X}_S} + n\lambda I$, $\boldsymbol{\alpha} = \{\alpha_j\}_{j=1}^{m}$ and $\lambda > 0$ is a regularisation parameter.*

*Proof.* Given $\hat{g}$, the preferential value function evaluated at $\hat{g}$ can be written as,

$$\hat{\nu}^{(p_I)}_{\mathbf{x}^{(\ell)},\mathbf{x}^{(r)},S}(\hat{g}) = \langle \hat{g}, \hat{\nu}^{(p_I)}_{\mathbf{x}^{(\ell)},\mathbf{x}^{(r)},S}\rangle_{\mathcal{H}_{k_E}} \tag{20}$$

$$= \langle \sum_{j=1}^{m} \alpha_j k_E((\mathbf{x}_j^{(\ell)}, \mathbf{x}_j^{(r)}), \cdot), \frac{1}{\sqrt{2}}\left(\hat{\mathcal{K}}(\mathbf{x}^{(\ell)}, S) \otimes \hat{\mathcal{K}}(\mathbf{x}^{(r)}, S) - \hat{\mathcal{K}}(\mathbf{x}^{(r)}, S) \otimes \hat{\mathcal{K}}(\mathbf{x}^{(\ell)}, S)\right)\rangle_{\mathcal{H}_{k_E}} \tag{21}$$

$$= \frac{1}{\sqrt{2}}\left\langle \sum_{j=1}^{m} \alpha_j k_E((\mathbf{x}_j^{(\ell)}, \mathbf{x}_j^{(r)}), \cdot), \hat{\mathcal{K}}(\mathbf{x}^{(\ell)}, S) \otimes \hat{\mathcal{K}}(\mathbf{x}^{(r)}, S)\right\rangle \tag{22}$$

$$- \frac{1}{\sqrt{2}}\left\langle \sum_{j=1}^{m} \alpha_j k_E((\mathbf{x}_j^{(\ell)}, \mathbf{x}_j^{(r)}), \cdot), \hat{\mathcal{K}}(\mathbf{x}^{(r)}, S) \otimes \hat{\mathcal{K}}(\mathbf{x}^{(\ell)}, S)\right\rangle \tag{23}$$

Now we focus on the first component, and rewrite:

$$\frac{1}{\sqrt{2}}\left\langle \sum_{j=1}^{m} \alpha_j k_E((\mathbf{x}_j^{(\ell)},\mathbf{x}_j^{(r)}),\cdot),\hat{\mathcal{K}}(\mathbf{x}^{(\ell)},S)\otimes\hat{\mathcal{K}}(\mathbf{x}^{(r)},S)\right\rangle = \sum_{j=1}^{m} A_j^{(1)} \tag{24}$$

and we continue to expand the terms,

$$A_j^{(1)} := \frac{1}{\sqrt{2}}\left\langle \alpha_j k_E((\mathbf{x}_j^{(\ell)},\mathbf{x}_j^{(r)}),\cdot),\hat{\mathcal{K}}(\mathbf{x}^{(\ell)},S)\otimes\hat{\mathcal{K}}(\mathbf{x}^{(r)},S)\right\rangle \tag{25}$$

$$= \frac{1}{\sqrt{2}}\left\langle \frac{\alpha_j}{\sqrt{2}}\left(k(\cdot,\mathbf{x}_j^{(\ell)})\otimes k(\cdot,\mathbf{x}_j^{(r)}) - k(\cdot,\mathbf{x}_j^{(r)})\otimes k(\cdot,\mathbf{x}_j^{(\ell)})\right),\hat{\mathcal{K}}(\mathbf{x}^{(\ell)},S)\otimes\hat{\mathcal{K}}(\mathbf{x}^{(r)},S)\right\rangle \tag{26}$$

$$= \frac{\alpha_j}{2}\left(\left\langle k(\cdot,\mathbf{x}_j^{(\ell)})\otimes k(\cdot,\mathbf{x}_j^{(r)}),\hat{\mathcal{K}}(\mathbf{x}^{(\ell)},S)\otimes\hat{\mathcal{K}}(\mathbf{x}^{(r)},S)\right\rangle - \left\langle k(\cdot,\mathbf{x}_j^{(r)})\otimes k(\cdot,\mathbf{x}_j^{(\ell)}),\hat{\mathcal{K}}(\mathbf{x}^{(\ell)},S)\otimes\hat{\mathcal{K}}(\mathbf{x}^{(r)},S)\right\rangle\right) \tag{27}$$

$$= \frac{\alpha_j}{2}\left(A_j^{(1,\ell)} - A_j^{(1,r)}\right) \tag{28}$$

We then note that

$$A_j^{(1,\ell)} := \left\langle k(\cdot,\mathbf{x}_j^{(\ell)}))\otimes k(\cdot,\mathbf{x}_j^{(r)}),\hat{\mathcal{K}}(\mathbf{x}^{(\ell)},S)\otimes\hat{\mathcal{K}}(\mathbf{x}^{(r)},S)\right\rangle \tag{29}$$

$$= \left\langle k(\cdot,\mathbf{x}_j^{(\ell)}),\hat{\mathcal{K}}(\mathbf{x}^{(\ell)},S)\right\rangle\left\langle k(\cdot,\mathbf{x}_j^{(r)}),\hat{\mathcal{K}}(\mathbf{x}^{(r)},S)\right\rangle \tag{30}$$

$$= k_S(\mathbf{x}_j^{(\ell)},\mathbf{x}^{(\ell)})\mathbf{K}_{\mathbf{x}_{jS}^{(\ell)},\mathbf{X}_S}(\mathbf{K}_{\mathbf{X}_S,\mathbf{X}_S}+n\lambda I)^{-1}\mathbf{K}_{\mathbf{X}_{S^c},\mathbf{x}_{S^c}^{(\ell)}} \tag{31}$$

$$\times k_S(\mathbf{x}_j^{(r)},\mathbf{x}^{(r)})\mathbf{K}_{\mathbf{x}_{jS}^{(r)},\mathbf{X}_S}(\mathbf{K}_{\mathbf{X}_S,\mathbf{X}_S}+n\lambda I)^{-1}\mathbf{K}_{\mathbf{X}_{S^c},\mathbf{x}_{S^c}^{(r)}} \tag{32}$$

$$= \Gamma(\mathbf{x}_{jS}^{(\ell)},\mathbf{x}_S^{(\ell)})\odot\Gamma(\mathbf{x}_{jS}^{(r)},\mathbf{x}_S^{(r)}) \tag{33}$$

To go from the second equation to the third equation in this paragraph, realise $k(\cdot,\mathbf{x}^{(\ell)}) = k_S(\cdot,\mathbf{x}_S^{(\ell)})\otimes k_{S^c}(\cdot,\mathbf{x}_{S^c}^{(\ell)})$ by product kernel assumption. In this case, we can rewrite $A_j^{(1)}$ as,

$$A_j^{(1)} = \frac{\alpha_j}{2}\left(\Gamma(\mathbf{x}_{jS}^{(\ell)},\mathbf{x}_S^{(\ell)})\odot\Gamma(\mathbf{x}_{jS}^{(r)},\mathbf{x}_S^{(r)}) - \Gamma(\mathbf{x}_{jS}^{(r)},\mathbf{x}_S^{(\ell)})\odot\Gamma(\mathbf{x}_{jS}^{(\ell)},\mathbf{x}_S^{(r)})\right) \tag{34}$$

Analogously, define $\sum A_j^{(2)}$ as the second component after the subtraction sign, by symmetry, we know

$$A_j^{(2)} = \frac{\alpha_j}{2}\left(\Gamma(\mathbf{x}_{jS}^{(\ell)},\mathbf{x}_S^{(r)})\odot\Gamma(\mathbf{x}_{jS}^{(r)},\mathbf{x}_S^{(\ell)}) - \Gamma(\mathbf{x}_{jS}^{(r)},\mathbf{x}_S^{(r)})\odot\Gamma(\mathbf{x}_{jS}^{(\ell)},\mathbf{x}_S^{(\ell)})\right) \tag{35}$$

by subtracting $A_j^{(1)}$ and $A_j^{(1)}$, we get the following:

$$\hat{\nu}_{\mathbf{x}^{(\ell)},\mathbf{x}^{(r)},S}^{(p_I)}(\hat{g}) = \sum_{j=1}^{m} \alpha_j\left(\Gamma(\mathbf{x}_{jS}^{(\ell)},\mathbf{x}_S^{(\ell)})\odot\Gamma(\mathbf{x}_{jS}^{(r)},\mathbf{x}_S^{(r)}) - \Gamma(\mathbf{x}_{jS}^{(r)},\mathbf{x}_S^{(\ell)})\odot\Gamma(\mathbf{x}_{jS}^{(\ell)},\mathbf{x}_S^{(r)})\right) \tag{36}$$

writing it in compact form, we arrive to our result,

$$= \boldsymbol{\alpha}^{\top}\left(\Gamma(\mathbf{X}_S^{(\ell)},\mathbf{x}_S^{(\ell)})\odot\Gamma(\mathbf{X}_S^{(r)},\mathbf{x}_S^{(r)}) - \Gamma(\mathbf{X}_S^{(\ell)},\mathbf{x}_S^{(r)})\odot\Gamma(\mathbf{X}_S^{(r)},\mathbf{x}_S^{(\ell)})\right) \tag{37}$$

$$\square$$

**Proposition 3.3** (Preferential value function for contexts). *Given a preference function $g_U \in \mathcal{H}_{k_E^U}$, denote $\Omega' = \{1,...,d'\}$, then the utility of context features $S' \subseteq \Omega'$ on $\{\mathbf{u},\mathbf{x}^{(\ell)},\mathbf{x}^{(r)}\}$ is measured by $\nu_{\mathbf{u},\mathbf{x}^{(\ell)},\mathbf{x}^{(r)},S'}^{(p_U)}(g_U) = \mathbb{E}[g_U(\{\mathbf{u}_{S'},U_{S'^c}\},\mathbf{x}^{(\ell)},\mathbf{x}^{(r)}) \mid U_{S'} = \mathbf{u}_{S'}]$ where the expectation is taken over the observational distribution of $U$. Now, given a test triplet $(\mathbf{u},\mathbf{x}^{(\ell)},\mathbf{x}^{(r)})$, if $\hat{g}_U = \sum_{j=1}^{m} \alpha_j k_E^U\left((\mathbf{u}_j,\mathbf{x}_j^{(\ell)},\mathbf{x}_j^{(r)}),\cdot\right)$, the non-parametric estimator is:*

$$\hat{\nu}_{\mathbf{u},\mathbf{x}^{(\ell)},\mathbf{x}^{(r)},S'}^{(p_U)}(\hat{g}_U) = \boldsymbol{\alpha}^{\top}\left(\left(\mathbf{K}_{\mathbf{U}_{S'},\mathbf{u}_{S'}}\odot\mathbf{K}_{\mathbf{U}_{S'^c},\mathbf{U}_{S'^c}}\left(\mathbf{K}_{\mathbf{U}_{S'},\mathbf{U}_{S'}}+m\lambda'I\right)^{-1}\mathbf{K}_{\mathbf{U}_{S'},\mathbf{u}_{S'}}\right)\odot\Xi_{\mathbf{x}^{(\ell)},\mathbf{x}^{(r)}}\right)$$

*where $\Xi_{\mathbf{x}^{(\ell)},\mathbf{x}^{(r)}} = \left(\mathbf{K}_{\mathbf{X}^{(\ell)},\mathbf{x}^{(\ell)}}\odot\mathbf{K}_{\mathbf{X}^{(r)},\mathbf{x}^{(r)}} - \mathbf{K}_{\mathbf{X}^{(r)},\mathbf{x}^{(\ell)}}\odot\mathbf{K}_{\mathbf{X}^{(\ell)},\mathbf{x}^{(r)}}\right).$*

*Proof.* Recall the feature map of the kernel $k_E^U$ takes the following form,

$$k_E^U\left((\mathbf{u}, \mathbf{x}^{(\ell)}, \mathbf{x}^{(r)}), \cdot\right) = k_u(\mathbf{u}, \cdot) \otimes k_E((\mathbf{x}^{(\ell)}, \mathbf{x}^{(r)}), \cdot) \tag{38}$$

Therefore we can express the preferential value function for context as,

$$\nu_{\mathbf{u}, \mathbf{x}^{(\ell)}, \mathbf{x}^{(r)}, S'}^{(p_U)}(g_U) = \mathbb{E}\left[g_U\left(\{\mathbf{u}_{S'}, U_{S^c}\}, \mathbf{x}^{(\ell)}, \mathbf{x}^{(r)}\right) \mid U_S = \mathbf{u}_{S'}\right] \tag{39}$$

$$= \left\langle g_U, \mathbb{E}\left[k_E^U\left((\{\mathbf{u}_{S'}, U_{S'^c}\}, \mathbf{x}^{(\ell)}, \mathbf{x}^{(r)}), \cdot\right) \mid U_{S'} = \mathbf{u}_{S'}\right]\right\rangle \tag{40}$$

$$= \left\langle g_U, \mathbb{E}\left[k_u(\{\mathbf{u}_{S'}, U_{S^c}\} \mid U_{S'} = \mathbf{u}_{S'}\right] \otimes k_E((\mathbf{x}^{(\ell)}, \mathbf{x}^{(r)}), \cdot)\right\rangle \tag{41}$$

$$= \left\langle g_U, k_{u_{S'}}(\mathbf{u}_{S'}) \otimes \mu_{U_{S'^c} \mid U_{S'} = \mathbf{u}_{S'}} \otimes k_E((\mathbf{x}^{(\ell)}, \mathbf{x}^{(r)}), \cdot)\right\rangle \tag{42}$$

The remaining steps are analogous to [12, Prop.2]. To obtain the empirical estimation, we first replace the conditional mean embedding $\mu_{U_{S'^c} \mid U_{S'} = \mathbf{u}_{S'}}$ with its empirical estimate and replace $g_U$ with $\hat{g}_U = \sum_{j=1}^m \alpha_j k_E^U\left((\mathbf{u}_j, \mathbf{x}_j^{(\ell)}, \mathbf{x}_j^{(r)}), \cdot\right)$. Now the empirical estimator has the following form,

$$\hat{\nu}_{\mathbf{u}, \mathbf{x}^{(\ell)}, \mathbf{x}^{(r)}, S'}^{(p_U)}(\hat{g}_U) = \left\langle \sum_{j=1}^m \alpha_j k_E^U\left((\mathbf{u}_j, \mathbf{x}_j^{(\ell)}, \mathbf{x}_j^{(r)}), \cdot\right), k_{u_{S'}}(\mathbf{u}_{S'}) \otimes \hat{\mu}_{U_{S'^c} \mid U_{S'} = \mathbf{u}_{S'}} \otimes k_E((\mathbf{x}^{(\ell)}, \mathbf{x}^{(r)}), \cdot)\right\rangle \tag{43}$$

$$= \sum_{j=1}^m \alpha_j \left\langle k_u(\mathbf{u}_j, \cdot) \otimes k_E((\mathbf{x}_j^{(\ell)}, \mathbf{x}_j^{(r)}), \cdot), k_{u_{S'}}(\mathbf{u}_{S'}) \otimes \hat{\mu}_{U_{S'^c} \mid U_{S'} = \mathbf{u}_{S'}} \otimes k_E((\mathbf{x}^{(\ell)}, \mathbf{x}^{(r)}), \cdot)\right\rangle \tag{44}$$

$$= \sum_{j=1}^m \alpha_j \left\langle k_u(\mathbf{u}_j, \cdot), k_{u_S}(\mathbf{u}_S) \otimes \hat{\mu}_{U_{S'^c} \mid U_S = \mathbf{u}_S}\right\rangle k_E\left((\mathbf{x}_j^{(\ell)}, \mathbf{x}_j^{(r)}), (\mathbf{x}^{(\ell)}, \mathbf{x}^{(r)})\right) \tag{45}$$

Now write everything in terms of matrices,

$$= \boldsymbol{\alpha}^\top \left(\left(\mathbf{K}_{\mathbf{U}_{S'}, \mathbf{u}_{S'}} \odot \mathbf{K}_{\mathbf{U}_{S'^c}, \mathbf{U}_{S'^c}}\left(\mathbf{K}_{\mathbf{U}_{S'}, \mathbf{U}_{S'}} + m\lambda' I\right)^{-1} \mathbf{K}_{\mathbf{U}_{S'}, \mathbf{u}_{S'}}\right) \odot \Xi_{\mathbf{x}^{(\ell)}, \mathbf{x}^{(r)}}\right) \tag{46}$$

where $\Xi_{\mathbf{x}^{(\ell)}, \mathbf{x}^{(r)}} = \left(\mathbf{K}_{\mathbf{X}^{(\ell)}, \mathbf{x}^{(\ell)}} \odot \mathbf{K}_{\mathbf{X}^{(r)}, \mathbf{x}^{(r)}} - \mathbf{K}_{\mathbf{X}^{(r)}, \mathbf{x}^{(\ell)}} \odot \mathbf{K}_{\mathbf{X}^{(\ell)}, \mathbf{x}^{(r)}}\right)$. $\qquad\square$