# OpenReview forum: "Explaining Preferences with Shapley Values"
_NeurIPS.cc/2022/Conference — NeurIPS 2022 Accept_

### Official Review · Reviewer_eoAJ · 2022-07-06

**Rating:** 6
**Confidence:** 4
**Soundness:** 3 good
**Presentation:** 4 excellent
**Contribution:** 2 fair

**Summary:**

The paper adapts the SHAP value estimation to the object-ranking task of the preference learning setting. Thereby, the authors present a novel way of computing SHAP values which the authors call PREF-SHAP. The authors present and evaluate the efficacy of PREF-SHAP with one synthetic dataset and three (plus one in the appendix) real-world ranking datasets. Moreover, the authors present their PREF-SHAP methodology formally by presenting a value function for the object ranking task of preference learning. The authors formally prove their methodology under the assumption that the ranking model under investigation is a kernel-based Generalised Preference Model. The proofs can be found in the appendix. Moreover, the authors publish their implementation online for open access.

**Questions:**

# Questions
- You frame your Pref-SHAP method to be the solution for all Preference learning/ranking machine learning models. However, you primarily apply RKHS-SHAP (or any SHAP estimation approach) to the object-ranking task on your mentioned kernel-based Generalised Preference Models.
	- Can you outline in what way your method is applicable to other types of preference models ($g(.)$) or tasks?
	- Why did you focus solely on kernel-based Generalised Preference Models? The sentence "While the ... " (lines 196 to 199) is not very informative for this big reduction of the problem.
- Is it possible to calculate and add the ground-truth Shapley values for your synthetic dataset and a "perfect model"?

# Suggestions
- Specify your method's main limitations and future work potential.
- To save some space, I would move some of the Pref-SHAP plots of section 4 into the appendix, as they are quite repetitive and take up a lot of space that could be better used for a thorough discussion of the limitations of your work or on a different kind of evaluation. I would keep the Pokemon and/or Tennis examples. These are great.
- There exists a paper from econometrics that first made the connection of calculating Shapley Values via the least-squares formulation. (Also mentioned by Lundberg and Lee in their SHAP paper). In line 143 of your manuscript, the proper citation could be added: Charnes, A., Golany, B., Keane, M., & Rousseau, J. (1987). Extremal Principle Solutions of Games in Characteristic Function Form: Core, Chebychev and Shapley Value Generalizations. In J. P. Ancot, A. J. H. Hallett, J. K. Sengupta, & G. K. Kadekodi (Eds.), Advanced Studies in Theoretical and Applied Econometrics. Econometrics of Planning and Efficiency (Vol. 11, pp. 123–133). Springer Netherlands. https://doi.org/10.1007/978-94-009-3677-5_7
- I suggest, adding a different focus to your evaluations: How do users interpret your explanations? How does RKHS-SHAP compare against regular KernelSHAP on the same setting (value function)

**Limitations:**

# Limitations
- The authors do not specify the limitations of their work.
- There is no discussion about the ethical implications of this work apart from the motivation. As I do not see big ethical concerns with such XAI works (rather the opposite) I don't think that this is a problem.

- The main limitation of the work is the more narrow application domain of object-ranking using a specific model type (kernel-based Generalised Preference Models). This is not clearly specified in the beginning.

- The evaluation of the method is done in a way that is clearly favoring the newly proposed method and no comparison is done to existing approximations (mainly KernelSHAP) in the same value-function setting (see the point in the "Quality" section).

- The proposed Pref-SHAP method is evaluated on a functional level without conducting any human experiments. For this work, a human experiment is not essential, as the work is rather foundational. However, as it is one of the first XAI + Preference Learning works, it is very interesting to see how people would interpret the output of Pref-SHAP, as it is unintuitive to do so even for a person who is quite familiar with SHAP.

**Strengths And Weaknesses:**

# Originality
The proposed Pref-SHAP is a natural extension of the already existing SHAP explanation framework. Pref-SHAP mainly builds on already existing work of Shapley Value approximation. The presented work relies heavily on a newly introduced RKHS-SHAP approximation method, which in itself is very new and, as of now, is still under review. Moreover, the work focuses heavily on a newly proposed model type of kernel-based Generalised Preference Models and the theoretical results are solely assuming such a model. Hence, the originality of the proposed work is rather incremental. (Application of SHAP to a new problem domain and reduction of problem space to analyze the solution formally)

That said, the presented work is to the best of my knowledge the first work linking a subset of preference learning and Shapley values explanations.

# Quality
In general, the quality of the work is high. The method is presented formally. The conducted empirical datasets are well selected and illustrate how SHAP on ranking functions can be used. As there exists no SOTA and, thus, it is sufficient to compare against a sensible baseline. The theoretical analysis is straightforward and interesting when the model function is kernel-based.

I understand the main contribution of the proposed work to be the value function in Definition 3.1 and that you show the results of evaluating the value function on a specific model instantiation (kernel-based Generalised Preference Models). The value function follows quite naturally from applying SHAP to the ranking domain and is presented very well. However, the authors do not give a valid motivation for why they chose to only focus on these specific kernel-based model functions. The only mention of this is the sentence: "While the ... " (lines 196 to 199), which does not include proper reasoning. Propositions 3.1 to 3.3 follow quite naturally from instantiating the model function $g(.)$ with a kernel-based GPM. Still, the theoretical results are interesting. Especially by framing the SHAP approximation problem as a kernel and utilizing RKHS (one of the main contribution of RKHS-SHAP, as I understand the paper).

I see a problem with the evaluation of PREF-SHAP and the chosen baseline. Of course, it is interesting to see what would happen if you were to concatenate the features of two instances and apply KernelSHAP rather naively. However, I argue that it is pretty obvious that KernelSHAP would not really work there, because it assumes feature independence, and the features are the same, and, thus, are dependent. A better comparison would be to group the same features of two instances together and remove both features simultaneously. Of course sampling from the conditional distribution is more problematic (solved by RKHS-SHAP, I figure) but in the removing-with-marginal case (after Covert et al. 2021, [28] in your work) this would be no problem and straightforward. I would be very interested, in why you chose to use these kernel-based GPMs and would be more interested in a thorough investigation of common Shapley value approximation techniques for different models and model types with your new value function (Definition 3.1).

# Clarity
In general, I enjoy the presentation of the work.
- I like the motivation and introduction giving a good overview into the field of preference learning for scholars not familiar with it. The same is true for the Shapley section. However, I suggest being more specific about what part of preference learning you are addressing (mainly object-ranking).
- I like the discussion of choosing marginal vs. conditional distribution for "feature removal".
- In particular, I like Table 1: This gives a good overview of the aforementioned preference learning / SHAP description
- Figure 1 is a great and small illustration of how the synthetic dataset was created and for what purpose it was implemented.
- In general, the selection of empirical test datasets suits the work greatly. The Pokémon dataset is clear to understand. The Tennis matches are a good example of a "real" real-world dataset and the focus on Djokovic's losses is a humorous and interesting illustration of your PREF-SHAP method
- The writing of the paper is good and it flows well. (Some hiccups along the way and the Propositions are a bit under-discussed)

### Minor Clarity Remarks:
- Coming from a Feature Importance background, I had problems with the formal notation of denoting the item index in the superscript in braces (l and r: $\boldsymbol{x}^{(l)}, \boldsymbol{x}^{(r)}$) as this representation is sometimes used to denote the feature column of the input space. Make this clear again in your paragraph introducing your notation.
- All Propositions (3.1 to 3.3) are hard to understand at first (though easier when following the proofs in the appendix). I suggest adding a better natural text description to the formulation to make it clearer.
- The contribution statement should be more specific and classify the main contribution more precisely (focus on kernel-based Generalised Preference Models)

- Minor Remarks:
	- Appendix A: I don't think that you wanted to say that your algorithm is "embarrassingly" parallel (probably you meant "embracing").
	- Proposition 3.3 (Line 223-224) is overflowing horizontally. (the same is true for some expressions in the appendix ... there this is no problem)
	- Sentence "An explicit..." (line 14-17) is very hard to read. The i.e. part through me off. Split this maybe in two.

# Significance
The significance of the work is high. Preference learning is a big part of machine learning research. Moreover, explainability and trustworthiness are also extremely important. Explainability is especially important for ranking problems, as search engines or recommender systems influence people on a daily basis. Being able to explain such systems is an important research question.

To the best of my knowledge, there exists no similar work of Shapley Value + any Preference Learning. As such, an application of the SHAP method to Preference Learning is a significant contribution to the community.

---

> ### Author Response · Authors · 2022-08-01
> **Response**
>
> Thank you for your time and effort in reviewing the paper. We reply to your questions and comments here:
>
> *Q1: Can you outline in what way your method is applicable to other types of preference models $g(\cdot)$ or tasks?*
>
> A1: While our proposed nonparametric estimator requires $g$ to live in an RKHS, our proposed value function is general and works for any skew-symmetric preference model. If one wants to model $g$ with a different model, such as a deep network, then one could appeal to methods discussed in Frye et al. 2020 to estimate our proposed value function, and proceed with the WLS approach in obtaining SVs as in KernelSHAP. We see our main contribution lies in devising a preference learning specific value function, and providing an effective way of estimating them that does not require solving a more difficult task of  conditional density estimation.
>
> - Frye et al. 2020 Shapley explainability on the data manifold
>
> ---
>
> *Q2: Why did you focus solely on kernel-based Generalised Preference Models? The sentence "While the ... " (lines 196 to 199) is not very informative for this big reduction of the problem.*
>
> A2: As replied above, we see our contribution in studying explainability under the context of preference learning. This has not been studied before and we devised an appropriate value function. There are numerous ways one could estimate the preference model, as well as the value function. We chose to focus on kernel-based generalised preference models because it allows us to utilise the recently proposed RKHS-SHAP to obtain closed form estimation of the value function, without estimating conditional densities, which are more difficult problems to solve than estimating conditional expectations. There are also convincing theoretical results about GPM as well given that the kernel used is universal with respect to the space of preference models.
>
> ---
>
> *Q3: Is it possible to calculate and add the ground-truth Shapley values for your synthetic dataset and a "perfect model"?*
>
> A3:  Thank you for this great suggestion – this will indeed be a helpful addition to the synthetic experiment, we will carefully consider calculating ground-truth Shapley values to better illustrate the significance of the results.
>
> We thank the reviewers for their suggestions and minor remarks, we will incorporate them into the paper.

---

> > ### Comment · Reviewer_eoAJ · 2022-08-09
> > **Rebuttal Discussion**
> >
> > Thank you for your response!
> > Your responses A1 to A2 are clear and help me understand your contribution better.
> >
> > However, you did not address one of my criticism. I feel like your baseline you compare against is quite unfair for classical SHAP.
> > Maybe I am missing an obvious point, but can you elaborate, why you are not comparing against a "standard" SHAP estimation approach (e.g. KernelSHAP) on a concatenation of features and remove the same features only in conjunction with each other. Wouldn't that make more sense than removing the features independently from each other in this concatenation?
> >
> > This is also, why I feel that comparing against ground-truth values (thank you for A3 for this) is vital for your work.

---

> > > ### Author Response · Authors · 2022-08-09
> > > **Response**
> > >
> > > Thank you for your question!
> > >
> > > The specific comparison done in appendix B is precisely meant to illustrate the importance of redefining the value function in order to make it suited for preferential data, i.e. to remove the features in conjunction with each other as the reviewer suggests -- and to assign Shapley values to the pair of features rather than to each player's feature individually (nb. individual Shapley values may also be interesting quantities in some contexts, cf. our response A6 to reviewer wsLb -- but they would answer a different type of explainability questions). However, an approach with a redefined value function is not quite the standard SHAP approach on concatenated data and our paper is the first, to the best of our knowledge, to give a formal treatment of explaining preferences and to address such questions. Essentially, the point that Shapley values need to be appropriately defined for preferential data (and that one cannot blindly run existing tools) is one of the main takeaways of our paper, in addition to our contribution in contrasting explaining preferences to explaining utility models.
> > > We hope this clarifies all reviewer's concerns.
> > > Thanks!

---

### Official Review · Reviewer_wsLb · 2022-07-09

**Rating:** 5
**Confidence:** 4
**Soundness:** 3 good
**Presentation:** 3 good
**Contribution:** 2 fair

**Summary:**

This paper proposes a method to explain attributions of inputs in a preference model. Specifically, the authors apply Shapley values to the preference model. To this end, the authors design the utility function for Shapley values on generalized preference models based on kernel functions. The proposed method can estimate attributions of both inputs and context variables.

**Questions:**

- I am confused about the claim that computing two Shapley values for the same feature in $x^{(l)}$ and $x^{(r)}$ leads to inconsistency. Although $x^{(l)}$ and $x^{(r)}$ consist of the same features, their feature values are different. Therefore, each feature value can be assigned a Shapley value. A simple example is that the Shapley value of $x_i^{(l)}=-x_i^{(r)}$.
- Some references for tables are incorrect. “Table 5” in Line289 should be “Table 2”, and “Table 4” in Line 308 should be “Table 3”.


**Limitations:**

The authors do not address the limitations of their work.

**Strengths And Weaknesses:**

[Strengths]
+ The paper is well-motivated and well-structured.
+ The authors do not directly apply Shapley values to preference models, but consider properties of preference models and propose an efficient way to compute Shapley values.


[Weaknesses]
- Although the proposed Pref-SHAP uses kernel functions to compute the value function $v$, its essence is actually the same as SHAP. The only difference is the implementation/computation of the value function, i.e., the model output when only given a subset of input variables. Some benefits of Pref-SHAP over SHAP on UPM actually come from advantages of GPM, rather than the explanation method.
- What is the advantage of computing $v$ by using kernel functions over directly computing the expected probability of the model? If using kernel functions is more efficient, then it would be better to compare the computational cost of the two implementations.
- In the third row of Table 2 for the candidate $x^{(l)}$, why not only modify $x^{(l)}$ in the computation of Shapley values without changing $x^{(r)}$?
- The comparison of local explanations in Figure 3 seems unfair. I think that one_hot$(c_i)$ should remain unchanged when computing $v(S)$ on GPM and UPM. In this way, the clusters of pairs are constant and explanations are restricted in the specified condition.
- For experiments on realistic datasets, how to judge which explanations are correct? For example, in Figure 6, both Pref-SHAP and SHAP for UPM show a chaotic pattern in beehive plots. Besides, differences in results between Pref-SHAP and SHAP for UPM may not stem from the explanation methods, but come from the difference between models, because UPM cannot perform well for unrankable tasks and may lead to strange explanations. It would be better if the authors compare the proposed method with other explanation methods, e.g., Integrate Gradient, on the same model.

---

> ### Author Response · Authors · 2022-08-01
> **Response Part 1 of 2**
>
> *C1: Although the proposed Pref-SHAP uses kernel functions to compute the value function $v$, its essence is actually the same as SHAP. The only difference is the implementation/computation of the value function, i.e., the model output when only given a subset of input variables. Some benefits of Pref-SHAP over SHAP on UPM actually come from advantages of GPM, rather than the explanation method.*
>
> A1: While SHAP provides an efficient way to compute Shapley values using weighted least square approach, it assumes one has access to the outcome of the value functions. As such, another route in Shapley value based explainability research is to devise new or improve estimations of the value function. Our contribution here lies in the latter where we studied a new explainability problem for preference learning, and propose an appropriate value function for it, along with an approach to estimating it.
>
> Could you please also clarify what benefits of Pref-SHAP over SHAP on UPM comes from GPM?
>
> ---
> *Q2: What is the advantage of computing by using kernel functions over directly computing the expected probability of the model? If using kernel functions is more efficient, then it would be better to compare the computational cost of the two implementations.*
>
> A2: Computing value function using kernel methods allows one to obtain close form solutions in terms of matrix vector multiplication, where one could then apply a variety of large scale kernel methods, and compute the quantity within $\mathcal{O}(n\sqrt{n})$ time [1]. If we do not use kernels, then the estimation of conditional density $p(X_{S^C} \mid X_S=\mathbf{X}_S)$ for all coalitions $S$ is required to compute the expectation, and that estimation problem might be more difficult than the original learning problem itself, and more costly [2].
>
> The discussion of computational advantage in using kernel functions for estimating value functions over neural methods have been discussed thoroughly in Chau et al (2022). The key contribution here is to study explainability for preference models since it is an unexplored but practical field. Our proposed value functions would still work even if we do not appeal to kernel methods.
>
> - [1] Rudi et al. 2017: FALKON: An optimal large scale kernel method
>
> - [2] Yeh et al. 2022: Threading the Needle of On and Off-Manifold Value Functions for Shapley Explanations
>
> ---
>
> *Q3: In the third row of Table 2 for the candidate $x^{(l)}$, why not only modify $x^{(l)}$  in the computation of Shapley values without changing $x^{(r)}$?*
>
> A3: If we only modify $x^{(l)}$ while keeping ONE particular $x^{(r)}$ fixed, then we are essentially asking “which item features contributed most to $x^{(l)}$’s match with this particular $x^{(r)}$”, instead of the question we are trying to address:“which item features contribute most to $x^{(l)}$’s matches”. They are different quantities.
>
> ---
>
> *Q4: For experiments on realistic datasets, how to judge which explanations are correct? For example, in Figure 6, both Pref-SHAP and SHAP for UPM show a chaotic pattern in beehive plots. Besides, differences in results between Pref-SHAP and SHAP for UPM may not stem from the explanation methods but come from the difference between models, because UPM cannot perform well for unrankable tasks and may lead to strange explanations. It would be better if the authors compare the proposed method with other explanation methods, e.g., Integrate Gradient, on the same model.*
>
> A4: Judging an explanation being “correct” is a very hard problem in itself since explainability is an unsupervised problem. We believe the comparison of explanation methods should happen at an axiomatic level, in that case, we can be more certain what propertiesour explanation will have. For example, Integrated Gradients (IG) was shown to theoretically approach Aummann-Shapley value as shown in [Chen et al. 2019], a fundamentally different concept to Shapey value. This difference leads to IG failing to satisfy some desirable feature attribution axioms that Shapley values do satisfy. For example, when feature i and j contributed equally to the function f across all coalitions S, i.e. $\nu_f(\{i\}\cup S) = \nu_f(\{j\}\cup S)$ with $\nu_f$ the value function defined with respect with $f$, IG do not necessarily returns the same attribution score to features i and j, but SVs would. Moreover, when feature i does not contribute to the function f at all, the attribution score from IG will not always be 0 while Shapley value based approach would. See examples from this article [1] for further reference.
>
> - [1] https://towardsdatascience.com/limitations-of-integrated-gradients-for-feature-attribution-ca2a50e7d269
> - [Chen et al. 2019] Explaining Models by Propagating Shapley values

---

> > ### Author Response · Authors · 2022-08-02
> > **Response Part 2 of 2**
> >
> > *Q6: I am confused about the claim that computing two Shapley values for the same feature in $x^{(l)}$ and $x^{(r)}$ leads to inconsistency. Although $x^{(l)}$ and $x^{(r)}$ consist of the same features, their feature values are different. Therefore, each feature value can be assigned a Shapley value. A simple example is that the Shapley value of $x^{(l)} = -x^{(r)}$.*
> >
> > A6: If one believes that the specific order between left players and right players carries additional information – if, for example, left always stands for “home team”, and right always stands for “away team”, then it might make sense to concatenate the two sets of features and obtain an explanation as you suggested, with separate Shapley values, where one can ask a different type of questions, e.g. “how relevant is this feature for the home team?”. However, in the general case when there is no specific meaning to this order (the case we consider in this paper), computing separate Shapley values will lead to difficulties for two reasons: (1) there is no principled way to aggregate the two different Shapley values for $x^{(l)}$ and $x^{(r)}$, as shown in the example in appendix B, (2) Comparing Shapley values for a specific feature (e.g. height) for the left player to that of the right player, we see that they do not actually satisfy the appropriate symmetry constraints – i.e. Shapley value for the “left height” at $(x^{(1)}, x^{(2)})$ need not be the same as the Shapley value for the “right height” at $(x^{(2)}, x^{(1)})$, i.e. explanations change purely due to the arbitrary order of players. This is the inconsistency which we alluded to in the manuscript, and it will be elaborated further in the final version. Our approach does not have such drawbacks: we are interested to know which feature, observed for both players, contributed most to the comparison between them and hence we define a value function that allows us to quantify the utility when such a feature is masked for both players, instead of treating two observations of the same feature separately.
> >
> > ---
> >
> > *C7: The comparison of local explanations in Figure 3 seems unfair. I think that one_hot should remain unchanged when computing $\nu(S)$ on GPM and UPM. In this way, the clusters of pairs are constant and explanations are restricted in the specified condition.*
> >
> > A7: One-hot encoding is unchanged when running GPM and UPM – i.e. they are run on the same data. We emphasise that figure 3 is just a plot of Shapley values restricted to matches between cluster A and B, while the actual estimation procedure is done over all observations, just like figure 2.

---

### Official Review · Reviewer_TiH3 · 2022-07-09

**Rating:** 8
**Confidence:** 3
**Soundness:** 3 good
**Presentation:** 4 excellent
**Contribution:** 4 excellent

**Summary:**

In the setting of explainable artificial intelligence, this paper investigates the challenging problem of explaining predictions made by preference models without strong rankability assumptions. To this point, one cannot exploit standard explainability tools for utility functions. So, this study instead considers the recent (possibly contextured) Generalized Preference Model proposed by Chau et al. (2022) that exploits kernels for capturing the likelihood function on pairs of items. By coupling this model with the paradigm of Shapley values, the authors derive a preferential value function for dueling items. Based on the existence of Rietz representation of the corresponding functional, this value function admits an elegant closed-form that can be computed efficiently. Experiments performed on a wide range of preference tasks corroborate the benefits of this Pref-SHAP approach, especially in comparison with a naive application of SHAP in pairwise preference explanation.


**Questions:**

No real questions, but see the comment above about the runtime complexity.


**Limitations:**

The Generalized Preference Model examined in this paper for inferring explanations makes no assumption about item rankability and takes only a few assumptions about the non-parametric function $g$ used in the model and its kernelization. The authors go even further by inferring explanations in the contextual setting. So, I did not find any real limitations in this framework.


**Strengths And Weaknesses:**

Overall the paper is very well-written and well-motivated. As many technical tools are introduced for deriving, in an efficient way, relevant explanations for general preference models, the paper is very dense. But all concepts are introduced parsimoniously, with a pedagogical effort made for explaining difficult parts. As far as I could check, the technical results look sound. Finally, the experiments performed on various tasks clearly highlight the practical utility of Pref-SHAP. In a nutshell, this is a good paper that shall pave the way for future research on explaining predictive preferences.

I found no real weaknesses in this paper. Just one point: it would be informative for the reader to give some bounds on the runtime complexity for estimating the value functions derived in Propositions 3.2 & 3.3.

---

> ### Author Response · Authors · 2022-08-01
> **Response**
>
> **Clarification on run time complexity**
>
> Thank you for your suggestion. For completeness, we will include a discussion on complexity here, and include it to the camera ready version later.
>
> When computing the GPM, since it is fundamentally a kernel ridge regression model, there are various large scale kernel approximation algorithms available. In this work, we chose to use FALKON[1], a Nyström approximation based preconditioner for conjugate gradient descent to find the regression solution in $\mathcal{O}(n\sqrt{n})$ time, where $n$ is the number of samples.
>
> When computing the value function for GPM, the challenge comes from computing the conditional mean embedding estimators, which in naive case, would require $\mathcal{O}(n\sqrt{n})$ complexity. However, one could again appeal to FALKON, and the estimation reverts back to $\mathcal{O}(n\sqrt{n})$ again.
>
> In practice, we found that when explaining larger datasets i.e. $n\sim 10^5$, it only took around 5 minutes on V100 cards to calculate all the Shapley values for all features for the entire dataset.
>
> [1]: FALKON: An Optimal Large Scale Kernel Method

---

### Official Review · Reviewer_jJ4D · 2022-07-11

**Rating:** 6
**Confidence:** 3
**Soundness:** 3 good
**Presentation:** 3 good
**Contribution:** 2 fair

**Summary:**

The paper seeks to `explain' the outcomes of binary contests (whether between two tennis players or elements of a consumer's choice set).  By imposing restrictions on the class of preference orderings considered, the paper is able to derive a ``closed form expression of the value function'' underlying the Shapley value, reducing the computational cost of determining the Shapley value.

**Questions:**

1. I would like to know what the generalization from UPM to GPM allows, in terms of utility theory.  I'm not familiar with the SOTA in this area, but wonder, for example, how GPM relates to e.g. Deaton and Muellbauer's `almost ideal demand system'.

1. line 193's ``inconsistent explanations'' makes me wonder what is being assumed about cross-partial elasticities (in the language of economic theory): is it assumed that features do not interact (i.e. neither substitutes nor complements)?  If so, that is a very strong assumption.

**Limitations:**

No concerns.

**Strengths And Weaknesses:**

**Originality**

The paper seems original to me, although in a somewhat complex way:
- a standard argument for Shapley value's use is that it is model agnostic.  From this point of view, knowing the underlying model defeats some of its motivation.
- utility functions (cardinal, ordinal, stochastic), preference orderings and the relationships between them have been studied for decades within the economics literature, as have (to a lesser extent) `contest success functions'.  The paper does not give indications of understanding that, perhaps making it seem more original than it is.

**Quality**

No concerns.

**Clarity**

Well presented.
- line 40: it could make sense to explain what the authors have in mind by ``to explain''.
- is the "unrankable" problem the same one that led to the use of stochastic utility in economic theory (q.v. work by e.g. Pattanaik, Barbera)?
- line 91: the standard approach in economic theory is to see a `good' as a complete description, thus including what are called `context variables' here.  It would be useful if the authors commented on why they do _not_ follow that approach.

**Significance**

Theoretically, I feel that the paper's detachment from the extensive microeconomic theory literature limits it.  (This said, I have not seen the question of applying Shapley values applied to consumer models.)

Thus, I needed to be convinced by the Experiments.  While I understand that they are `toy examples', it may still be that they trivialize the material (e.g. who cares which Pokeman wins a match, or Djokovic's performance on clay?).  Personally, I found myself wanting to know e.g. what this can say about inter-state wars (e.g. using https://correlatesofwar.org/data-sets/COW-war/dyadic-inter-state-war-dataset-1).  This both seems orders of magnitude more important, and might inform a topical issue, Russia's invasion of Ukraine.

---

> ### Author Response · Authors · 2022-08-01
> **Response**
>
> Thank you for your time and effort in reviewing the paper. We answer your questions and comments here:
>
> ---
>
> *Q1: I would like to know what the generalization from UPM to GPM allows, in terms of utility theory. I'm not familiar with the SOTA in this area, but wonder, for example, how GPM relates to e.g. Deaton and Muellbauer's `almost ideal demand system'.*
>
> A1: To the best of our knowledge, utility models often map items of interest to a scalar and pairwise preferences are then derived by comparing them. This imposes a strict ordering between items and as Chau et al 2022 showed, cannot be used to model complex, more realistic preferences that often exhibit, e.g. cyclic structure. GPM bypasses this scalar structural assumption and directly models the pairwise preference function nonparametrically, thus can learn more general preference structures than UPM.
>
> We are not familiar with the ‘almost ideal demand system’ model but it seems to be a specific demand model with a parametric form relating price and utility level with a cost function. We do not believe this is relevant to our work on explaining pairwise preference.
>
> - Chau et al. 2022: Learning Inconsistent Preferences with Gaussian Processes
>
> ---
>
> *C2: line 193's ``inconsistent explanations'' makes me wonder what is being assumed about cross-partial elasticities (in the language of economic theory): is it assumed that features do not interact (i.e. neither substitutes nor complements)? If so, that is a very strong assumption.
> A2: We do not make such an assumption in our work. This assumption will lead to trivial solutions to the Shapley value computation, since now $\nu({i} \cup S) = \nu({i})$ for all coalition S if features do not interact, where $\nu$ is the usual value function.*
>
> ---
>
> *C3: Personally, I found myself wanting to know e.g. what this can say about inter-state wars (e.g. using https://correlatesofwar.org/data-sets/COW-war/dyadic-inter-state-war-dataset-1). This both seems orders of magnitude more important, and might inform a topical issue, Russia's invasion of Ukraine.*
>
> A3: Thank you for the suggestion. We believe the focus of our contribution is methodological. Therefore, the aim was to demonstrate different aspects of our method using a variety of suitable datasets with appropriate formats. A potential application like the one the reviewer suggested would require careful consideration and its importance would warrant the entirety of the focus of such applied work.
>
> ---
>
> *Q4: is the "unrankable" problem the same one that led to the use of stochastic utility in economic theory (q.v. work by e.g. Pattanaik, Barbera)?*
>
> A4: We do not see immediate connections to stochastic utility in economic theory, because the GPM model used does not model utility, but only the outcome of the comparison.
>
> Thank you for the suggestions, we have incorporated the feedback into our manuscript.

---

> > ### Comment · Reviewer_jJ4D · 2022-08-09
> > **acknowledgement of rebuttal**
> >
> > Having re-read my comments, and the authors' reply, I do not feel myself more convinced than I was on first reading the manuscript.
> >
> > Above all, the motivation remains unclear to me:
> > - are we using a model-agnostic explainability technique on a non-black box model?
> > - as I understand it, the authors prefer to remove the transitivity assumption in the usual definition of 'rational' preferences over using stochastic utility.  I don't know this field well enough to understand why that is a better approach.

---

> > > ### Author Response · Authors · 2022-08-09
> > > **Responses**
> > >
> > > Q: are we using a model-agnostic explainability technique on a non-black box model?
> > >
> > > A: What we are providing in this paper is:
> > >
> > > 1) A novel general framework for explaining skew-symmetric preference models that need not assume transitivity.
> > > 2) Under this general framework, we chose to estimate the value functions specifically using RKHS methods. This need not be the case and there are many other methods to do something similar, check out Covert et al. 2022, Frye et al. 2020.
> > >
> > > [Covert et al. 2022] Explaining by Removing:A Unified Framework for Model Explanation
> > >
> > > [Frye et al. 2020] Shapley explainability on the data manifold
> > >
> > > ---
> > >
> > > Q: Preference over removal of transitivity assumption
> > >
> > > A: This approach is built upon a line of existing researches that study flexible modelling over preferences. In practice, total rankability of preferences are often too strong an assumption. There might be many reasons why some "noisy" preference do not conform to a single overall ranking. For example, it is well studied that cognitive biases often lead to inconsistent human preferences in behavioural economics (Tversky et al. 1992). We encourage the reviewer to see the work of Causer et al. 2005, Pahikkala et al. 2010, Waegeman et al. 2012, Chen et al. 2016, and Chau et al. 2022 on their motivation to consider intransitive relations.
> > >
> > > [Tversky et al 1992] Advances in prospect theory: Cumulative representation of uncertainty.
> > >
> > > [Causeur et al. 2005] A 2-dimensinoal extension of the Bradley-Terry model for paired comparisons.
> > >
> > > [Pahikkala et al. 2010] Learning intransitive reciprocal relations with kernel methods.
> > >
> > > [Waegeman et al. 2012] A kernel-based framework for learning graded relations from data.
> > >
> > > [Chen et al. 2016] Modeling intransitivity in matchup and comparison data.
> > >
> > > [Chau et al. 2022] Learning inconsistent preference with Gaussian Processes.

---

### Author Response · Authors · 2022-08-01
**General response**

We thank the reviewers for their helpful comments and suggestions, we sincerely believe they have significantly improved this work. We are happy that the reviewers found the work well-written, technically sound, and significant. We respond to each reviewer individually below.

---

### Meta-Review · Area_Chair_2p7U · 2022-08-27

**Recommendation:** Accept
**Confidence:** Certain

**Metareview:**

Overall, the opinion about this paper is quite positive, especially because of its novelty: It establishes the first connection between preference learning and explainability/Shapley. In terms of presentation and technical soundness, the paper seems to be convincing, too. A few critical points (e.g., regarding the evaluation) have been raised in the reviews, but they could essentially be resolved in the discussion. Another critical issue that came up in the final discussion is the following one: The authors learn a binary preference predicate g(X,Y) predicting the degree of preference of X over Y, though without any constraints. In particular, such a model may induce violations of transitivity in the sense that X>Y and Y>Z and Z>X. Such inconsistencies are debatable from a (normative) preference modeling point of view, although it's true that they can be observed in practice. In any case, they appear to be important from an EXPLAINABILITY point of view, as they might be confusing to the user. This point isn't addressed in the paper.

**Award:**

No

---

### Decision · Program_Chairs · 2022-09-14

Accept